# Context-Aware Contrastive Surrogates for Scaling-up Expensive Multiobjective Optimization

## Abstract

Addressing expensive multiobjective optimization problems (EMOPs) poses a significant challenge due to the high cost of objective evaluations. We propose FSMOEA, a scalable and efficient framework that enhances surrogate-assisted multiobjective evolutionary algorithms (SMOEAs) by introducing foresighted surrogate models. FSMOEA captures population-level context to improve surrogate prediction accuracy, leverages a low-dimensional latent space to accelerate evolutionary search, and employs lightweight models to reduce computational overhead. Designed for plug-and-play integration, the foresight model can be embedded into existing contrastive (i.e., classification- and relation-based) SMOEAs, improving performance on scaling-up EMOPs. We provide theoretical analysis that formalizes the benefits of population-aware representation and latent-space optimization. Extensive experiments on 107 benchmarks show that FSMOEA consistently outperforms state-of-the-art methods in both convergence speed and optimization quality. Source code is attached and will be available at *Linkxxx*.

## 1 Introduction

Multi-objective optimization problems (MOPs) arise in diverse domains such as neural architecture search (Zhou et al., 2023), deep learning (Chen & Kwok, 2022), multitask learning (Sener & Koltun, 2018), aerodynamic design (Jin et al., 2018), and drug discovery (Nicolaou & Brown, 2013), where practitioners seek to optimize multiple conflicting objectives simultaneously. Solving these problems yields a Pareto front (PF) — a set of trade-off solutions where no objective can be improved without degrading another (Cai et al., 2023). Gradient-free multiobjective evolutionary algorithms (MOEAs) have been extensively applied to these problems due to their population-based nature and robustness to non-convex, multimodal search spaces (Huang et al., 2024). These algorithms alternate between generating candidate solutions (using a generator), evaluating them (by a evaluator), and selecting the most promising ones (with a discriminator), gradually evolving towards a well-distributed approximation of the PF (Zhang et al., 2021). However, a key limitation of conventional MOEAs is their reliance on a large number of expensive objective function evaluations (Liu et al., 2022a), making them impractical for real-world scenarios involving high-fidelity simulations. This has motivated the development of surrogate-assisted MOEAs (SMOEAs), which approximate objective functions using cheaper predictive models such as Kriging (Song et al., 2021), radial basis function (Yu et al., 2019), support vector regression, or neural networks (Guo et al., 2021). These surrogates accelerate convergence while preserving solution quality (Li et al., 2022).

A principled way to address expensive black-box optimization is through Bayesian optimization (BO), which treats the objective as a random function and iteratively refines its belief over the function using Gaussian processes or other uncertainty-aware models (Xie et al., 2024; Tay et al., 2023). BO is data-efficient, balancing exploration and exploitation via acquisition functions such as expected improvement (EI) or upper confidence bound, and has seen significant success in single-objective settings (Ament et al., 2023). However, extending BO to expensive MOPs (EMOPs) is challenging due to the high-dimensional trade-off space and the difficulty of maintaining well-calibrated uncertainty estimates across all objectives (Lin et al., 2022b; Wei et al., 2024). SMOEAs can be viewed as a scalable, population-based counterpart to BO, enabling better exploration of large and complex search spaces through surrogate-guided evolutionary search (Zhou et al., 2024).

Existing SMOEAs for EMOPs fall into two broad categories: regression-based and contrastive (including classification-based and relation-based) SMOEAs. Regression-based methods directly model the objective values and use them to guide search and selection (Chugh et al., 2016; Knowles, 2006; Zhao & Zhang, 2023), but often suffer from modeling inaccuracies in high-dimensional or sparse-data regimes. Contrastive SMOEAs instead model pairwise performance relationships, e.g., which of two solutions is better, and leverage lightweight classifiers to perform surrogate selection (Yuan & Banzhaf, 2021; Hao et al., 2022; Zhang et al., 2022). This bypasses the need to predict exact objective values and is often more robust under data scarcity (Sonoda & Nakata, 2022). Despite their promise, contrastive SMOEAs face two critical challenges (Yang et al., 2023). First, existing models typically lack context-awareness — that is, they treat each comparison independently without considering the population-wide distribution. This limits their ability to generalize selection pressure across dynamic evolutionary landscapes. Second, scalability remains a bottleneck: performance degrades substantially as the problem dimensionality increases, hampering their applicability to scaling-up EMOPs (e.g., with many-objective and large-scale search space).

In this work, we propose a foresighted surrogate framework to address these issues. Our method introduces three key innovations: 1) In-context foresight: a context-aware head learns population-level embeddings to better capture selection dynamics; 2) Low-dimensional code space learning: a learned latent representation facilitates more efficient and generalizable comparisons; 3) Lightweight surrogate architecture: our method remains scalable and computationally efficient across problem sizes. These design choices collectively yield improved convergence, robustness, and computational efficiency on challenging scaling-up EMOPs.

## 2 RELATED WORK AND MOTIVATION

### 2.1 EXPENSIVE MULTIOBJECTIVE OPTIMIZATION

An MOP with $m$ objectives to be minimized is generally formulated as:

$$\min F(\mathbf{x}) = (f_1(\mathbf{x}), f_2(\mathbf{x}), \ldots, f_m(\mathbf{x}))^{\mathrm{T}}, \quad \text{s.t. } \mathbf{x} \in \Omega \tag{1}$$

where $\mathbf{x} = (x_1, x_2, \ldots, x_n)^{\mathrm{T}}$ is a candidate solution in an $n$-dimensional decision space $\Omega$, and $F(\mathbf{x})$ denotes a vector of $m$ potentially conflicting objective functions. The goal is to identify the Pareto set: a collection of non-dominated solutions that map to the PF in objective space. In computationally intensive settings, where each evaluation of the objective vector $F(\mathbf{x})$ incurs a significant cost, this task becomes markedly more challenging. Let $t_F$ denote the computational cost of a single evaluation of $F(\mathbf{x})$, and let $FE_{\max}$ be the maximum number of evaluations allowed under a fixed budget $T_{\text{budget}}$. We model this constraint as: $T_{\text{budget}} = t_F \times FE_{\max}$. This constraint motivates the development of strategies that prioritize high-utility evaluations and avoid wasteful exploration (Li et al., 2025). Please see Section E in the Appendix for more details of an MOP.

EMOPs are pervasive in domains like robotics, materials science, and automated machine learning, where simulation or experiment-driven evaluations dominate the runtime (Jin et al., 2018). Traditional MOEAs operate in an evaluation-hungry manner, relying on the sheer volume of function calls to ensure convergence. When $t_F$ is large, however, the allowable $FE_{\max}$ often drops by orders of magnitude — making naive MOEA strategies ineffective. This has led to a surge in interest around SMOEAs, where a learned model substitutes the true objective evaluator for most candidate solutions (Khaldi & Draa, 2024).

### 2.2 SURROGATE-ASSISTED MOEAS (SMOEAS)

To alleviate the cost of evaluating expensive objective functions, SMOEAs incorporate learned approximations (or surrogates) to filter and prioritize candidate solutions. These surrogates are integrated into the standard MOEA pipeline, which typically consists of a generator, evaluator, and selector (Liu et al., 2023). While the generator explores new regions of the search space using evolutionary operators such as crossover and mutation, the evaluator estimates objective values (or rankings) of the generated candidates, and the selector identifies the most promising solutions for survival and reproduction. In the SMOEA context, the evaluator is replaced or augmented by a surrogate model trained on a limited archive of truly evaluated solutions. This model acts as a proxy to the expensive function $F(\mathbf{x})$, providing fast but approximate predictions to guide the search. Only

a small subset of the most promising candidates, as estimated by the surrogate, are selected for real evaluation. This mirrors the role of acquisition functions in Bayesian optimization, which determine where to sample next based on model uncertainty and expected improvement. SMOEAs can be broadly categorized into three types based on the nature of the surrogate:

**Regression-based surrogates** learn a direct mapping from $\mathbf{x} \mapsto F(\mathbf{x})$ using models such as Kriging, support vector regression, radial basis functions, or neural networks (Si et al., 2023; Li et al., 2024b; Si et al., 2023; Gu et al., 2024). After prediction, a standard selection criterion (e.g., dominance, decomposition, or indicator-based) is applied to identify elite solutions. While effective under dense training data, regression models often struggle when data is sparse or high-dimensional, leading to unreliable estimates. **Classification-based surrogates** sidestep the need for precise function prediction by instead learning to classify solutions as promising or non-promising (Pan et al., 2018; Hao et al., 2021; Li et al., 2024a). This binary simplification is more robust under limited data and reduces the modeling complexity. Classifiers can be trained using pairwise comparisons or labels derived from environmental selection criteria. **Relation-based surrogates** further generalize classification by predicting relative rankings between pairs of solutions (Hao et al., 2022; Chen & Zhang, 2024; Hao et al., 2025). Rather than absolute labels, these models estimate which solution in a pair is likely superior, enabling fine-grained selection even when objective values are unknown or noisy.

A representative framework is shown in Fig.1, where the surrogate assists the evolutionary process by prioritizing candidates for real evaluation. As illustrated in Fig.2, these models are trained on previously evaluated solutions, refined iteratively, and queried during offspring generation to guide the evolutionary trajectory. Importantly, surrogate models must be computationally lightweight — their inference and update time must remain negligible compared to the cost of real evaluations.

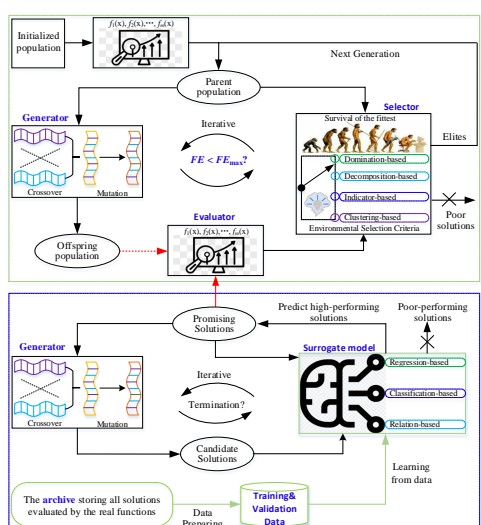

For regression models, the training data is often insufficient to span the high-dimensional search space, especially when only a few hundred evaluations are permissible. In contrast, classification and relation models exploit structural relationships among evaluated solutions, which can be more sample-efficient and robust to noise. Surrogate management is another critical component of SMOEAs: models must be regularly validated, retrained with new data, and dynamically adjusted to maintain reliability over time. Moreover, improper model guidance can cause premature convergence or population collapse — particularly when the surrogate's prediction errors are not well-calibrated. Finally, while surrogate models used in SMOEAs may appear functionally similar to those in BO, their usage context differs.

Figure 1: Overview of SMOEAs. The top depicts a standard MOEA loop with real evaluations, while the bottom illustrates how a learned surrogate pre-selects promising candidates for expensive evaluation, reducing computational cost.

fers. Bayesian multi-objective optimization (MOBO) typically employs Gaussian processes and acquisition functions to drive sample selection (Daulton et al., 2020; Konakovic Lukovic et al., 2020; Belakaria et al., 2019). While highly data-efficient, MOBO methods scale poorly in many-objective and high-dimensional settings due to computational bottlenecks in surrogate training and acquisition function optimization (Tu et al., 2022; Wang et al., 2023; Ozaki et al., 2024). In contrast, SMOEAs scale more naturally due to their population-based nature, implicit diversity maintenance, and parallel search capabilities.

## 2.3 INSIGHT AND MOTIVATIONS

Regression-based surrogates remain mainstream in high-dimensional optimization, yet they face two critical challenges in expensive settings: (i) dimensionality reduction often degrades the accuracy required for reliable regression, and (ii) data scarcity makes accurate function approximation infeasible. By contrast, contrastive surrogates (including classification- and relation-based models) are more tolerant to information loss, require fewer samples, and can exploit richer relational data

(e.g., $\mathcal{O}(N^2)$ pairwise labels from $N$ evaluations). These properties make them particularly suitable for SMOEAs. Despite these advantages, existing contrastive surrogates still exhibit limitations in large-scale or evolving populations. First, they lack population context-awareness: labels are derived from global criteria (e.g., dominance or decomposition), but predictions are made in isolation without modeling the surrounding population. As distributions drift, this mismatch leads to biased or inconsistent predictions (Li et al., 2024b). Second, scalability remains problematic: as objective and search spaces dimensionality grows, surrogate complexity scales linearly or worse, limiting responsiveness in iterative updates (Liu et al., 2024). Third, surrogate quality is tightly linked to search efficiency: poor discrimination in early generations can stall exploration, especially in high-dimensional landscapes where informative features are sparse. To overcome these issues, we propose a foresighted surrogate architecture that (i) integrates population context via a learned latent space, (ii) enables efficient low-dimensional search, and (iii) employs lightweight classifiers for scalability. This design improves surrogate accuracy, robustness, and overall efficiency in solving scaling-up EMOPs. For dimensionality reduction, while similar in spirit to PCA as studied in (Lin et al., 2022a; Gu et al., 2024), our method differs fundamentally by embedding population-aware context into the dimensionality reduction process. This allows the latent space to evolve with the search, making it directly useful for surrogate modeling and evolutionary guidance.

## 3 THE PROPOSED ALGORITHM

We propose FSMOEA, a foresight-enhanced SMOEA, which augments traditional contrastive SMOEAs with a foresight model $M_F$ to improve scalability and context-awareness in solving EMOPs. FSMOEA introduces a population-aware encoding-decoding mechanism via $M_F$, an autoencoder trained on the current population. As illustrated in Figure 2, $M_F$ consists of an encoder and decoder, with a hidden layer of size $k \ll n$, where $n$ is the dimensionality of the decision space. The encoder projects solutions into a compact latent space, while the decoder reconstructs them. Training minimizes the reconstruction loss (mean squared error) using standard backpropagation.

The autoencoder is trained exclusively on the current population $\mathcal{P}_t$ so that its representation reflects the geometry and distribution of solutions at that generation. This design explicitly captures the phenomenon of *population drift*, i.e., the gradual shift in neighborhood structure and decomposition-based scalarization values across generations. Once trained, the encoder serves as a frozen foresight head that captures the structural features of the current population. FSMOEA uses this head in two key ways: 1) to enhance surrogate predictions via context-aware representation, and 2) to conduct evolutionary search directly in the learned latent space, improving both efficiency and scalability. To ensure stable encoding under small population sizes, the foresight model employs a lightweight architecture aligned with the size of $\mathcal{P}_t$, together with mild regularization. This prevents overfitting while preserving the local geometric relations needed for downstream surrogate modeling. The latent dimension $k$ is selected to approximate the intrinsic dimensionality of $\mathcal{P}_t$; empirical analyses show that a small range of $k$ provides consistently robust representations.

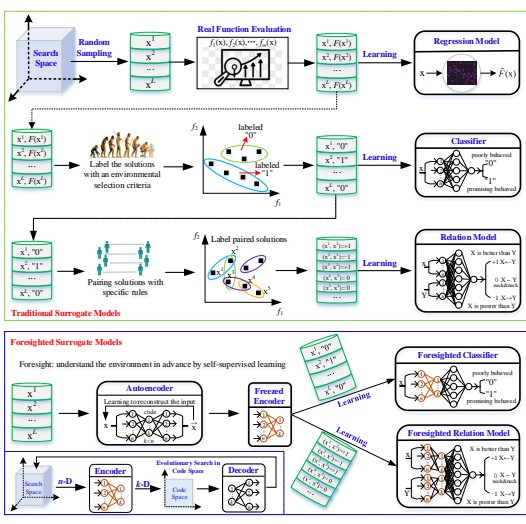

Figure 2: The basic framework of SMOEAs.

The high-level procedure proceeds as follows: 1) initialize a population $\mathcal{P}$ with $N$ solutions evaluated using the true objective function $F(\mathbf{x})$; 2) train the foresight model $M_F$ on $\mathcal{P}$; 3) perform surrogate-assisted search based on $M_F$ to generate an offspring population $O$; and 4) apply environmental selection on $\mathcal{P} \cup O$ to form the next generation. This process repeats until the evaluation budget $FE_{\max}$ is exhausted. The pseudocode and detailed description of the FSMOEA framework are provided in Appendix Section C. The computational overhead introduced by $M_F$ is modest: training a shallow autoencoder on $N$ samples of dimensionality $n$ requires $\mathcal{O}(Nk^2)$ operations,

and because the composite surrogate $M_{FS}$ receives $k$-dimensional inputs, its training complexity is significantly reduced compared to operating directly in the original $n$-dimensional space.

### 3.1 FORESIGHT SURROGATE-ASSISTED EVOLUTIONARY SEARCH FOR REPRODUCTION

Algorithm 2 in the appendix describes the surrogate-assisted reproduction mechanism. The key idea is to integrate the foresight encoder with an existing classifier- or relation-based surrogate model $M_S$, forming a composite surrogate $M_{FS}$. The encoder projects each solution into the latent space, serving as the fixed input layer for $M_S$. This reduces input dimensionality from $n$ to $k$, lowering training complexity and improving generalization. In FSMOEA, the autoencoder serves as one practical realization of the proposed population-aware dimensionality-reduction mechanism, but it is not the only possible choice. Any mapping that can both (i) compress high-dimensional decision variables into a compact latent representation and (ii) capture population-dependent structure may be used. We adopt a lightweight autoencoder primarily because it offers a flexible and computationally efficient way to obtain such context-aware embeddings together with an explicit decoder, which is convenient for latent-space evolutionary operations. The framework itself, however, remains agnostic to the specific dimensionality-reduction model employed.

We instantiate FSMOEA in two settings: FCSEA extends CSEA (Pan et al., 2018) by attaching the foresight encoder to its classifier, and FREMO extends REMO (Hao et al., 2022) using the same principle for relation modeling. In both cases, model training and surrogate management follow the structure of the original baselines. The foresight encoder is trained once per generation and then frozen during surrogate updates. Because $k \ll n$, the composite model $M_{FS}$ is lighter and faster to train. See Section I for sensitivity analysis of $k$.

FSMOEA performs evolutionary operations in the learned latent space. Given two parent solutions $x$ and $y$, their latent codes $c_x, c_y \in \mathbb{R}^k$ are combined via crossover and mutation to produce a new code $c$, which the decoder maps back to the original decision space to form a new candidate solution $z$. This latent-space search accelerates convergence by exploring a compact, structured subspace shaped by the current population. This design makes FSMOEA not only more scalable but also broadly compatible with existing classification- and relation-based SMOEAs. More implementation details are provided in Section C of the appendix.

FSMOEA introduces three core contributions. **Context-aware evaluation**: The foresight encoder encodes population-level information, enabling the surrogate to evaluate new candidates with richer contextual understanding. **Improved efficiency and scalability**: By operating in a reduced latent space, FSMOEA improves surrogate training speed and stability, particularly in high-dimensional settings. **Faster convergence**: Latent-space search improves the quality of generated candidates, leading to faster identification of Pareto-optimal solutions under tight evaluation budgets.

### 3.2 THEORETICAL ANALYSIS OF KEY COMPONENTS

This section provides theoretical justification for the two central design choices in FSMOEA: (1) the use of MLP-based autoencoders to obtain population-aware embeddings for surrogate modeling, and (2) the use of latent-space search to reduce sample complexity and accelerate convergence. For EMOPs, the performance of a solution $\mathbf{x}$ is often assessed relative to a population $\mathcal{P}_t$ under different dominance schemes. Pareto dominance can become ineffective in high dimensions due to a lack of discriminative power (Liu et al., 2022c;b). Decomposition-based dominance addresses this issue by evaluating solutions through scalarization functions tailored to each subproblem (He et al., 2017; Yuan et al., 2016). FSMOEA inherits from CSEA and REMO, which rely on such decomposition-guided strategies. In this setting, the relative quality of solutions depends on population-derived quantities such as the ideal point and neighborhood structure; therefore, as the population evolves, the dominance outcome itself changes. This section formalizes this dependence and explains how the proposed autoencoder-based embedding mitigates the resulting inconsistency.

**Population Drift and Decomposition-Based Dominance.** Let $w \in \mathbb{R}^m$ be a normalized weight vector associated with a subproblem. Given an objective vector $f(x) \in \mathbb{R}^m$, the widely used scalarization functions (as in CSEA and REMO) is:

$$g^{\text{PBI}}(x|w, z^*) = \frac{\langle f(x) - z^*, w \rangle}{\|w\|} + \theta \left\| f(x) - z^* - \frac{\langle f(x) - z^*, w \rangle}{\|w\|^2} w \right\|, \quad (2)$$

where $z^*$ is the ideal point and $\theta > 0$ is a penalty factor. Each criterion $C(x; \mathcal{P}_t)$ is computed relative to the current population, since both $z^*$ and several neighborhood-based components depend on $\mathcal{P}_t$. Thus, if $\mathcal{P}_t \neq \mathcal{P}_{t+1}$, we may have $C(x; \mathcal{P}_t) \neq C(x; \mathcal{P}_{t+1})$, even for fixed $x$.

**Population drift under decomposition.** If $z_t^* \neq z_{t+1}^*$, then $g^{\mathrm{PBI}}(x|w, z_t^*) \neq g^{\mathrm{PBI}}(x|w, z_{t+1}^*)$. Thus, decomposition-based dominance is inherently *population-dependent*. This population dependence is what we refer to as *population drift*: generation-to-generation variations in $\mathcal{P}_t$ induce corresponding shifts in decomposition scores, altering dominance outcomes and local neighborhoods. Since drift can occur even when objective vectors change only mildly, a surrogate that assumes fixed target labels becomes fundamentally inconsistent. A surrogate $M : \mathbb{R}^n \to \mathbb{R}$ is context-free if $M(x)$ is computed independently of $\mathcal{P}_t$. Suppose $M$ is trained on labels $y_i = C(x_i; \mathcal{P}_t)$. If $\mathcal{P}_t \neq \mathcal{P}_{t+1}$, then $C(x_i; \mathcal{P}_t) \neq C(x_i; \mathcal{P}_{t+1})$ while $M(x_i)$ remains fixed, introducing systematic bias. For example, a point $x$ may be non-dominated in $\mathcal{P}_t$ but dominated in $\mathcal{P}_{t+1}$, yet $M(x)$ continues to approximate the earlier label. This shows that context-free surrogates necessarily incur prediction bias when labels depend on a drifting population.

**Remedy via Context-Aware Embeddings.** FSMOEA addresses this issue by using an autoencoder $M_F = (E, D)$ trained directly on $\mathcal{P}_t$. The encoder $E_t$ provides population-aware codes $c_i = E_t(x_i)$ that reflect global structure in $\mathcal{P}_t$, enabling the surrogate $\tilde{M}$ to approximate $M(x; \mathcal{P}_t) = \tilde{M}(E_t(x))$, rather than mapping from raw decision vectors. As $E_t$ is retrained each generation, the surrogate input space adapts consistently with the evolving dominance relations. To control representation drift, the encoder is updated only when its validation reconstruction error deviates beyond a small threshold, ensuring that $E_t$ varies smoothly across generations. This corresponds to bounding the encoder drift parameter $\eta = \|E_t - E_{t-1}\|$, which appears explicitly in the error propagation analysis (Appendix D). Autoencoders are also theoretically preferable over linear projections such as PCA in this setting, because they can preserve nonlinear manifold structure and provide an explicit decoder $D$ enabling inverse mapping required for latent-space genetic operations. Variants such as VAEs introduce stochasticity in decoding, which is undesirable for deterministic reproduction.

**Latent-Space Fidelity and Smoothness.** If $D$ is $L_D$-Lipschitz with bounded reconstruction error $\epsilon$, and $F$ is $L_F$-Lipschitz, then for any latent codes $z_1, z_2$:

$$\|F(D(z_1)) - F(D(z_2))\| \leq L_F L_D \|z_1 - z_2\| + 2 L_F \epsilon.$$

Thus, smoothness in the latent space is transferred to the objective space up to controlled distortion. This bound clarifies the effect of dimensionality reduction: as long as the reconstruction error remains bounded and $D$ is sufficiently smooth, evolutionary operators in latent space induce reliable and interpretable variations in the decision space. The additive $2 L_F \epsilon$ term quantifies the theoretical tolerance of FSMOEA to imperfect reconstruction, which is especially important for high-dimensional EMOPs.

**Complexity Benefits.** Because latent-space optimization operates in $\mathbb{R}^k$ with $k \ll n$, the sample complexity required for surrogate training and the search-space volume explored per generation both decrease substantially. Combined with the Lipschitz-based distortion bound above, this shows that FSMOEA can reduce effective search complexity without sacrificing structural fidelity. This analysis shows that: (1) Population drift under decomposition-based dominance introduces inherent inconsistencies for context-free surrogates. (2) FSMOEA's autoencoder provides dynamic, population-aware embeddings that remain aligned with evolving dominance relations. (3) Latent-space search reduces sample complexity while preserving smoothness and fidelity. Detailed theoretical analysis, including formal definitions, lemmas, and proofs, is provided in Appendix D.

# 4 EXPERIMENTAL EVALUATION

We conduct comprehensive experiments to evaluate the effectiveness and scalability of the proposed FSMOEA framework, instantiated in two surrogate-assisted algorithms: FCSEA and FREMO. These are benchmarked against sixteen state-of-the-art methods, including regression-based (KRVEA (Chugh et al., 2016), SMSEGO (Ponweiser et al., 2008), EDNARMOEA (Guo et al., 2021), ADSAPSO (Lin et al., 2022a), LDSAF (Gu et al., 2024), SFADE (Horaguchi et al., 2025)), Bayesian-based (ABSAEA (Wang et al., 2020), ESBCEO (Bian et al., 2023),

DirHVEI (Zhao & Zhang, 2024), MORBO (Rashidi et al., 2024)), classification- and relation-based SMOEAs (CSEA (Pan et al., 2018), REMO (Hao et al., 2022), MCEAD (Sonoda & Nakata, 2022), MOL2SMEA (Si et al., 2025)). Two SMOEAs, i.e., EICMSSAEA (Wu et al., 2025) and RECMO Liu et al. (2025), specifically for constrained EMOPs are also included.

Table 1: Average IGD values of FCSEA, FREMO, and their ablated variants (FCSEA-V1, CSEA, FREMO-V1, REMO) on DTLZ1–7 and WFG1–9 problems with $m = 3$ and $N = 50$.

| Problems | $n$ | CSEA | FCSEA-V1 | FCSEA | REMO | FREMO-V1 | FREMO |
|---|---|---|---|---|---|---|---|
| DTLZ1 | 50 | 7.1977e+2(9.14e+1)+ | **6.9746e+2(8.47e+1)+** | 9.7583e+2(2.70e+2) | 9.4858e+2(2.79e+2)- | 6.8327e+2(1.20e+2)+ | **6.6910e+2(8.76e+1)** |
| | 100 | 1.8764e+3(1.13e+2)= | 1.8681e+3(1.62e+2)= | **1.6804e+3(8.97e+2)** | 1.8061e+3(1.74e+2)- | 1.8098e+3(1.65e+2)- | **1.7105e+3(9.32e+2)** |
| DTLZ2 | 50 | 1.5057e+0(2.11e-1)- | 1.2749e+0(1.95e-1)- | **5.2832e-1(6.31e-2)** | 1.1560e+0(1.98e-1)- | 1.2116e+0(1.68e-1)- | **5.4138e-1(1.24e-1)** |
| | 100 | 4.0946e+0(4.02e-1)- | 3.9648e+0(4.03e-1)- | **6.2574e-1(1.48e-1)** | 3.8417e+0(4.81e-1)- | 3.7357e+0(3.31e-1)- | **7.4766e-1(3.18e-1)** |
| DTLZ3 | 50 | 3.0932e+3(8.32e+2)- | **2.0737e+3(2.17e+2)=** | 2.2088e+3(1.76e+2) | 2.7929e+3(9.82e+2)- | **2.0227e+3(2.19e+2)+** | 2.0644e+3(2.42e+2) |
| | 100 | 6.1106e+3(4.81e+2)- | 6.0580e+3(3.72e+2)- | **5.6272e+3(2.92e+3)** | 5.7370e+3(3.45e+2)- | 5.8136e+3(4.01e+2)- | **5.2558e+3(3.48e+3)** |
| DTLZ4 | 50 | 1.3299e+0(1.94e-1)- | 1.1843e+0(1.97e-1)- | **9.2653e-1(1.55e-1)** | 1.3046e+0(1.64e-1)- | 1.1298e+0(1.46e-1)- | **9.9888e-1(1.45e-1)** |
| | 100 | 3.7547e+0(3.90e-1)- | 3.5439e+0(2.81e-1)- | **9.9549e-1(1.11e-1)** | 3.7898e+0(3.62e-1)- | 3.6973e+0(4.21e-1)- | **1.0238e+0(2.63e-1)** |
| DTLZ5 | 50 | 1.4034e+0(2.20e-1)- | 1.2371e+0(2.07e-1)- | **4.0903e-1(1.28e-1)** | 1.1644e+0(1.79e-1)- | 1.1465e+0(2.06e-1)- | **4.2520e-1(2.08e-1)** |
| | 100 | 3.8830e+0(3.97e-1)- | 3.8055e+0(3.86e-1)- | **5.7915e-1(3.52e-1)** | 3.8369e+0(4.17e-1)- | 3.7947e+0(4.52e-1)- | **4.9651e-1(1.73e-1)** |
| DTLZ6 | 50 | 4.1080e+1(6.96e-1)- | 3.6590e+1(1.12e+0)- | **3.6340e+1(1.23e+0)** | 4.0330e+1(9.37e-1)- | 3.6642e+1(1.44e+0)= | **3.6451e+1(1.38e+0)** |
| | 100 | 8.5634e+1(9.10e-1)- | 8.0986e+1(1.11e+0)- | **7.9495e+1(1.68e+0)** | 8.5179e+1(8.88e-1)- | 8.1296e+1(1.52e+0)- | **8.0799e+1(1.79e+0)** |
| DTLZ7 | 50 | 8.0987e+0(9.94e-1)- | 4.4684e+0(9.46e-1)= | **4.4665e+0(8.17e-1)** | 7.2926e+0(8.66e-1)- | 3.5760e+0(8.24e-1)- | **3.3692e+0(8.00e-1)** |
| | 100 | 9.2832e+0(6.77e-1)= | **6.1156e+0(6.57e-1)=** | 6.1247e+0(7.47e-1) | 8.8404e+0(6.81e-1)- | **5.7992e+0(7.20e-1)-** | 5.8163e+0(4.60e-1) |
| WFG1 | 50 | 2.1504e+0(1.08e-1)- | 1.6301e+0(1.03e-1)- | **1.5278e+0(6.62e-2)+** | 1.9785e+0(1.58e-1)- | **1.5590e+0(3.76e-2)=** | 1.5670e+0(4.12e-2) |
| | 100 | 2.0790e+0(1.21e-1)- | **1.6334e+0(6.95e-2)-** | 1.5806e+0(1.28e-1)+ | 1.9181e+0(1.41e-1)- | 1.5780e+0(3.32e-2)= | **1.5680e+0(3.65e-2)** |
| WFG2 | 50 | 6.1959e-1(3.45e-2)- | **6.0090e-1(3.62e-2)+** | 6.6454e-1(4.29e-2) | 6.4678e-1(6.73e-2)= | 6.5416e-1(4.55e-2)- | **6.1975e-1(4.62e-2)** |
| | 100 | 6.7467e-1(2.02e-2)= | 6.7985e-1(2.33e-2)= | **6.6723e-1(5.03e-2)** | 6.9451e-1(4.37e-2)- | **6.4816e-1(4.91e-2)=** | 6.6985e-1(4.17e-2) |
| WFG3 | 50 | 7.0072e-1(3.62e-2)- | 6.8423e-1(3.38e-2)- | **5.5687e-1(2.74e-2)** | 6.7492e-1(4.78e-2)- | 6.6887e-1(4.10e-2)- | **5.6397e-1(3.74e-2)** |
| | 100 | 7.4720e-1(3.56e-2)- | 7.6099e-1(3.17e-2)- | **5.5762e-1(3.55e-2)** | 7.4968e-1(2.23e-2)- | 7.5056e-1(2.81e-2)- | **5.5750e-1(3.26e-2)** |
| WFG4 | 50 | 4.8438e-1(2.42e-2)+ | **4.7344e-1(2.02e-2)+** | 5.2181e-1(3.01e-2) | 5.0351e-1(3.46e-2)- | 4.7057e-1(2.46e-2)= | **4.6128e-1(1.85e-2)** |
| | 100 | 5.1278e-1(2.28e-2)= | **5.0605e-1(1.60e-2)+** | 5.3294e-1(4.16e-2) | 5.2928e-1(3.24e-2)- | 5.0115e-1(1.50e-2)= | **4.9666e-1(1.75e-2)** |
| WFG5 | 50 | 7.4924e-1(1.72e-2)- | 6.5753e-1(1.86e-2)- | **6.2561e-1(3.47e-2)+** | 7.3966e-1(1.81e-2)- | 6.4695e-1(3.83e-2)= | **6.4070e-1(2.63e-2)** |
| | 100 | 7.6078e-1(9.56e-3)- | **7.0552e-1(2.06e-2)+** | 7.0740e-1(2.69e-2)+ | 7.6569e-1(1.28e-2)- | 6.9550e-1(2.40e-2)= | **6.9454e-1(2.51e-2)** |
| WFG6 | 50 | 8.2959e-1(2.50e-2)- | 8.1146e-1(2.50e-2)- | **8.0017e-1(2.39e-2)** | 8.4198e-1(4.05e-2)- | 8.2419e-1(2.99e-2)- | **8.0372e-1(2.71e-2)** |
| | 100 | 8.9024e-1(1.72e-2)- | 8.7077e-1(2.26e-2)- | **8.2709e-1(2.20e-2)** | 8.9234e-1(2.54e-2)- | 8.7694e-1(2.36e-2)- | **8.2400e-1(2.66e-2)** |
| WFG7 | 50 | 6.7276e-1(2.49e-2)- | 6.5438e-1(2.30e-2)- | **6.0914e-1(1.35e-2)** | 6.6321e-1(3.00e-2)- | 6.6139e-1(2.50e-2)- | **6.0562e-1(1.39e-2)** |
| | 100 | 7.0070e-1(1.85e-2)- | 6.8507e-1(1.94e-2)- | **6.2251e-1(1.45e-2)** | 6.9108e-1(1.70e-2)- | 6.8881e-1(2.28e-2)- | **6.2302e-1(1.75e-2)** |
| WFG8 | 50 | 7.2910e-1(3.42e-2)- | **7.0166e-1(2.33e-2)+** | 7.1390e-1(1.77e-2) | 7.2182e-1(2.30e-2)- | **7.0529e-1(2.71e-2)=** | 7.0808e-1(3.26e-2) |
| | 100 | 7.6027e-1(2.47e-2)- | 7.2506e-1(2.52e-2)- | **7.1474e-1(2.06e-2)** | 7.3887e-1(2.34e-2)- | 7.2698e-1(2.82e-2)- | **7.0999e-1(1.96e-2)** |
| WFG9 | 50 | 8.5295e-1(6.49e-2)- | 8.5317e-1(5.60e-2)- | **7.6220e-1(4.32e-2)** | 8.4950e-1(7.12e-2)- | 8.5860e-1(6.68e-2)- | **7.6224e-1(4.62e-2)** |
| | 100 | 9.2945e-1(4.19e-2)- | 9.1105e-1(3.75e-2)- | **7.7577e-1(5.19e-2)** | 9.1149e-1(6.94e-2)= | 9.2901e-1(6.03e-2)- | **7.6596e-1(6.46e-2)** |
| +/-/= | | vs. FCSEA: 3/26/3 | vs. FCSEA: 5/21/6 | —— | vs. FREMO: 0/28/4 | vs. FREMO: 2/23/7 | —— |

## 4.1 EXPERIMENTAL SETUP

We evaluate the selected algorithms on eight widely used test suites: DTLZ (Deb et al., 2005), WFG (Huband et al., 2006), MaF (Cheng et al., 2017), LSMOP (Cheng et al., 2016), MLDMP (Li et al., 2017), MPDMP (Köppen & Yoshida, 2007), real-world SMOP (Tian et al., 2019), and TREE (He et al., 2020), comprising 112 benchmark instances with diverse numbers of objectives and decision variables. DTLZ and WFG are classical synthetic benchmarks widely adopted in multi-objective optimization. MaF and LSMOP are designed for many-objective and large-scale scenarios, respectively. MLDMP and MPDMP represent real-world multi-line and multi-point distance minimization tasks. The real-world SMOP suite includes neural network training (MOP-NN), feature selection (MOP-FS), and signal reconstruction (MOP-SR). TREE consists of industrial-scale voltage transformer calibration problems. This benchmark selection reflects standard EMO evaluation practices, encompassing a broad range of synthetic and real-world problems across multi-, many-objective, and high-dimensional settings. Performance is measured using the inverted generational distance (IGD), $IGD^+$, and Hypervolume (HV) metrics, assessing convergence and diversity.

Each algorithm is executed over 30 independent runs per instance. All implementations use recommended parameters; the evaluation budget is fixed at 500 function evaluations with a population size of 50. For FSMOEA, the latent dimension is set to $k = 10$, while FCSEA and FREMO inherit all other settings from their respective baselines (CSEA and REMO). Statistical significance is determined using the Wilcoxon rank-sum test at the 0.05 level. In all result tables, symbols "+", "-", and "=" denote cases where FCSEA or FREMO significantly underperform, outperform, or match the baseline, respectively. Best scores are highlighted in bold. All source codes were implemented on the PlatEMO (Tian et al., 2017), and all experiments were conducted on a personal computer equipped with an Intel Core i5-10505 CPU (3.2 GHz) and 24 GB of RAM. For clarity, we emphasize that our experimental setup was designed to be fair and stringent; detailed justifications on problem selection, evaluation budget, and efficiency are provided in Section G of the Appendix.

## 4.2 EFFECTIVENESS AND COMPONENT-WISE ABLATION

To isolate the impact of FSMOEA's core components—the foresight head and latent-space search—we conduct ablation studies on DTLZ1–7 and WFG1–9. We define two ablated variants: (1) FCSEA-V1, which retains the foresight head but performs search in the original space, and (2) FREMO-V1, analogously defined for FREMO. These are compared against their baselines (CSEA, REMO) and full FSMOEA variants. Results (Table 1; see Appendix for full versions) show that both foresight-enhanced variants (FCSEA, FREMO) consistently outperform their ablated counterparts, particularly in higher-dimensional decision spaces ($n \in \{50, 100\}$). While FCSEA-V1 and FREMO-V1 provide modest gains over CSEA and REMO, they fall short of the full FSMOEA variants—indicating that the latent representation is critical for scaling to large $n$. The foresight head contributes significant performance gains by embedding context-awareness into the surrogate model, while latent-space search accelerates convergence and enhances sample efficiency.

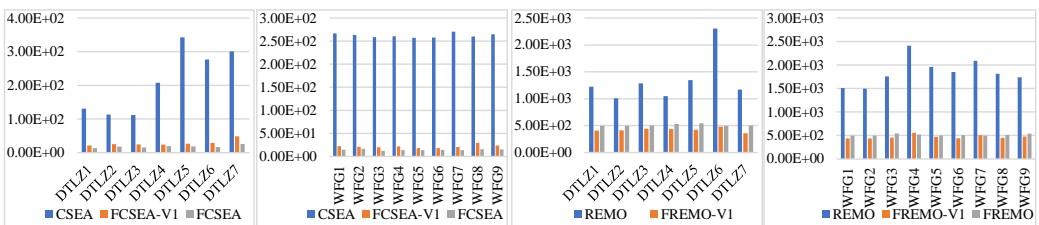

Figure 3: Illustration of the average actual running time (as y-axis: the unit is Seconds) of FCSEA, FREMO and their variants in solving DTLZ and WFG problems ($m = 3, n = 100$).

## 4.3 RUNTIME ANALYSIS AND COMPUTATIONAL EFFICIENCY

We evaluate the practical runtime of FCSEA, FREMO, and their ablated variants to assess computational efficiency, particularly under high-dimensional settings. Fig 3 reports the average runtime (in seconds) across 30 independent runs on DTLZ1–7 and WFG1–9 test problems with $m = 3$ and $n = 100$. Notably, FCSEA exhibits runtime performance comparable to its variant FCSEA-V1, indicating that the addition of the foresight head introduces negligible overhead. More importantly, both FCSEA and FREMO achieve up to an order-of-magnitude speedup over their baselines, CSEA and REMO, respectively. This performance gap is consistent across all benchmark functions. The observed efficiency gains stem from two key factors in FSMOEA. First, the use of an MLP-based foresight head compresses input dimensionality from $n$ to $k$ (with $k \ll n$), significantly reducing the number of parameters in the downstream classifier or relational surrogate. Second, the encoder is frozen during surrogate training, allowing for rapid, deterministic embeddings and eliminating back-propagation overhead within the latent model. Together, these design choices enable faster inference and lower memory consumption, contributing to both runtime efficiency and improved scalability in large-scale EMOPs. Overall, FSMOEA's architectural simplicity, combined with latent-space search and lightweight surrogates, enables efficient optimization with tight evaluation and time budgets.

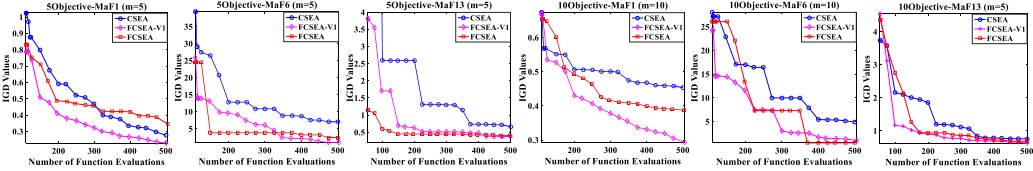

Figure 4: Convergence curves of FCSEA, FREMO and their variants on selected MaF benchmark problems (MaF1, MaF6, and MaF13) with varying objective dimensionality.

## 4.4 SCALABILITY WITH RESPECT TO OBJECTIVES AND VARIABLES

We further evaluate scalability from two orthogonal perspectives: objective dimensionality and variable dimensionality. For objective scalability, we assess FCSEA on the MaF1–13 suite under many-objective settings ($m \in \{5, 10\}$). Convergence curves for selected functions (MaF1, MaF6, MaF13) are shown in Figure 4. FCSEA demonstrates faster convergence and better final IGD scores than

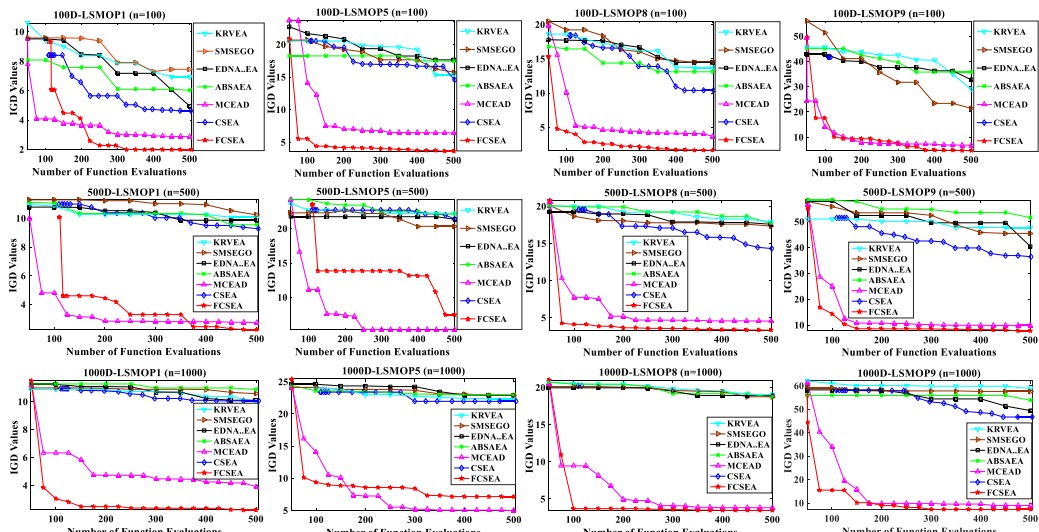

Figure 5: Convergence curves of FCSEA and its six competitors on selected LSMOP benchmarks (LSMOP1, LSMOP5, LSMOP8, and LSMOP9) with varying variable dimensionality.

Table 2: Average $IGD^+$ results of FCSEA and its six competitors in solving 3-objective 100-dimensional DTLZ and WFG problems with $N$=50 and $FE_{max} = 500$.

| Problem | MORBO | DirHVEI | SFADE | LDSAF | ADSAPSO | MOL2SMEA | FCSEA |
|---------|-------|---------|-------|-------|---------|----------|-------|
| DTLZ2 | 3.8257e+0(3.19e-1) | 6.1744e+0 (1.68e-2) | **5.0768e-1(6.07e-2)** | 6.4925e-1(1.50e-1) | 3.6555e+0 (6.04e-1) | 1.5651e+0(1.85e-1) | 5.0800e-1(7.68e-2) |
| DTLZ4 | 3.4314e+0 (3.84e-1) | 6.4274e+0(2.09e-1) | 6.1700e-1(2.43e-2) | 7.1070e-1(1.80e-1) | 4.7245e+0(1.30e+0) | 1.3316e+0(1.15e-1) | **5.5030e-1(3.65e-2)** |
| DTLZ7 | 9.9169e+0 (6.86e-1) | 9.8915e+0(2.10e-1) | 6.2735e+0(5.89e-1) | 9.0743e+0(5.34e-1) | 7.3988e+0(1.06e+0) | 1.0389e+1(5.56e-1) | **5.7365e+0(3.10e-1)** |
| WFG1 | 2.2018e+0(7.77e-2) | 2.1343e+0(2.99e-2) | **1.6034e+0(2.37e-2)** | 2.0756e+0(5.76e-2) | 2.1984e+0(3.92e-2) | 1.9965e+0 (1.14e-1) | 1.6151e+0(3.38e-2) |
| WFG5 | 7.7329e-1(1.62e-2) | 7.4307e-1(6.90e-3) | 3.1820e-1(4.53e-2) | 3.5816e-1(3.81e-2) | 7.0728e-1(5.91e-2) | 7.8875e-1 (1.80e-2) | **2.9240e-1(4.63e-2)** |
| WFG8 | 7.1172e-1(1.86e-2) | 7.4256e-1(7.89e-3) | 6.6978e-1(3.76e-2) | 6.9288e-1(3.32e-2) | 7.9785e-1(3.44e-2) | 5.9404e-1(2.94e-2) | **5.1502e-1(2.46e-2)** |

both its variants (FCSEA-V1 and CSEA), confirming that foresighted surrogates enhance generalization even in many-objective scenarios. For variable scalability, we evaluate FCSEA on the LSMOP suite with high-dimensional decision spaces ($n \in \{100, 500, 1000\}$), comparing it against six strong competitors. As shown in Figure 6, FCSEA significantly outperforms regression-based (KRVEA, SMSEGO, EDNARMOEA) and Bayesian-based (ABSAEA) surrogates. It also surpasses classification-based MCEAD and its own baseline CSEA in most cases. The combination of lightweight latent representations and population-aware surrogate modeling enables FSMOEA to scale to large $n$ without compromising performance or stability.

To further validate the scalability of our method on large-scale EMOPs, we additionally compare FCSEA with six algorithms specifically designed for high-dimensional EMOPs: MORBO, DirHVEI, SFADE, LDSAF, ADSAPSO, and MOL2SMEA. Table 2 reports the $IGD^+$ values on 100-dimensional 3-objective DTLZ and WFG problems under a tight evaluation budget. FCSEA consistently attains competitive or superior performance, confirming the effectiveness of the proposed population-aware latent representation in high-dimensional settings.

## 4.5 PERFORMANCE ON REAL-WORLD PROBLEMS

To assess the practical effectiveness of FCSEA in solving real-world EMOPs, we evaluate it on ten diverse benchmark problems, including MLDMP, MPDMP), MOP_NN), MOP_FS, MOP_SR, and five TREE problems. We compare FCSEA against six competitive algorithms: KRVEA, LDSAF, ABSAEA, ESBCEO, MCEAD, and CSEA. Each algorithm is given the same strict evaluation budget of 500 function evaluations. Table 3 reports the average HV results across 30 runs. FCSEA achieves comparable or superior performance on MLDMP and MPDMP, where all methods operate in low-dimensional decision spaces ($n = 2$). More notably, FCSEA outperforms all competitors on the remaining high-dimensional real-world problems, particularly excelling in large-scale tasks like MOP_NN, MOP_PO, and MOP_SR. The most significant advantage of FCSEA is observed on the TREE suite. While all other algorithms fail to find any feasible solutions within the evaluation budget—resulting in 'NaN' HV scores—FCSEA successfully discovers valid, high-quality

Figure 6: Average rankings of FCSEA and its ten competitors based on the IGD+ results.

Table 3: Average HV results of FCSEA and its six competitors in solving real-world EMOPs with $N$=50 and $FE_{max} = 500$, NaN denotes failure to find any feasible solution.

| Problems | $(m, n)$ | KRVEA | LDSAF | ABSAEA | ESBCEO | MCEAD | CSEA | FCSEA |
|---|---|---|---|---|---|---|---|---|
| MLDMP | (3, 2) | 6.732e-1(3.43e-2) | 1.920e-1(3.00e-1) | 6.956e-1(2.85e-2) | 8.155e-2(1.82e-1) | 4.619e-1(1.22e-1) | 2.207e-1(1.43e-1) | **8.276e-1(4.05e-3)** |
| MPDMP | (4, 2) | 2.577e-1(4.87e-3) | 5.315e-2(1.19e-1) | **2.778e-1(1.73e-3)** | 1.047e-1(9.49e-2) | 1.456e-1(2.86e-2) | 3.938e-2(5.21e-2) | 2.727e-1(1.87e-2) |
| MOP_NN | (2, 321) | 7.734e-2(6.58e-4) | 8.235e-2(7.15e-4) | 7.698e-2(5.89e-4) | 2.953e-1(2.06e-2) | 8.174e-2(5.93e-4) | 7.791e-2(3.25e-4) | **3.429e-1(9.88e-3)** |
| MOP_PO | (2, 1000) | 9.131e-2(2.58e-5) | 9.156e-2(1.36e-4) | 9.131e-2(4.16e-5) | 9.127e-2(5.56e-5) | 9.141e-2(8.63e-5) | 9.136e-2(4.69e-5) | **9.162e-2(1.64e-4)** |
| MOP_SR | (2, 1024) | 0.000e+0(0.0e+0) | 0.000e+0(0.0e+0) | 0.000e+0(0.0e+0) | 6.992e-2(2.16e-2) | 0.000e+0(0.0e+0) | 0.000e+0(0.0e+0) | **8.975e-2(5.24e-3)** |
| TREE1 | (2, 300) | NaN(NaN) | NaN(NaN) | NaN(NaN) | NaN(NaN) | NaN(NaN) | 6.366e-1(1.71e-2) | **7.909e-1(5.03e-2)** |
| TREE2 | (2, 300) | NaN(NaN) | NaN(NaN) | NaN(NaN) | NaN(NaN) | NaN(NaN) | NaN(NaN) | **7.636e-1(3.80e-2)** |
| TREE3 | (2, 600) | NaN(NaN) | NaN(NaN) | NaN(NaN) | NaN(NaN) | NaN(NaN) | NaN(NaN) | **8.727e-1(1.30e-2)** |
| TREE4 | (2, 600) | NaN(NaN) | NaN(NaN) | NaN(NaN) | NaN(NaN) | NaN(NaN) | NaN(NaN) | **8.773e-1(8.97e-2)** |
| TREE5 | (2, 600) | NaN(NaN) | NaN(NaN) | NaN(NaN) | NaN(NaN) | NaN(NaN) | NaN(NaN) | **8.556e-1(8.84e-2)** |

solutions across all five TREE problems. This suggests that FCSEA not only generalizes well to large-scale real-world scenarios but also exhibits strong robustness and sample efficiency in highly constrained, evaluation-limited settings. The ability to maintain convergence and feasibility under such constraints highlights the practical superiority of the FSMOEA framework.

TREE is a real-world constrained task, where only solutions that satisfy all constraints are considered feasible. In Table 3, "NaN" indicates that the algorithm failed to discover any feasible solution within the evaluation budget, leaving the final population empty and the HV metric undefined. Although FSMOEA does not incorporate explicit constraint-handling techniques, it successfully locates feasible solutions on TREE (The HV results of both the EICMSSAEA and RECMO in solving these five TREE problems are also NaN.). This demonstrates that its context-aware modeling and latent-space search accelerate convergence toward the feasible region in large-scale spaces.

**Ablation study**. We emphasize that our framework is not tied to autoencoders; any dimensionality-reduction module that can capture population-dependent structure and provide a reversible mapping is compatible with FSMOEA. Thus, we replaced the autoencoder with PCA and VAE, producing the variants CSEA-PCA and CSEA-VAE. As shown in Table 4, both variants remain competitive but consistently underperform FCSEA. This confirms that while FSMOEA does not rely on autoencoders, the nonlinear yet deterministic embeddings produced by a lightweight autoencoder provide a more stable and population-aligned latent space, thereby enhancing surrogate accuracy and search efficiency.

Table 4: Ablation study: Average $IGD^+$ results of FCSEA and its dimensionality-reduction variants (CSEA-PCA and CSEA-VAE) on 3-objective 100-dimensional DTLZ and WFG problems. Lower values indicate better performance.

| Problem | CSEA-PCA | CSEA-VAE | FCSEA |
|---|---|---|---|
| DTLZ2 | 6.12e-1(9.1e-2) | 5.65e-1(8.2e-2) | **5.08e-1(7.6e-2)** |
| DTLZ4 | 6.82e-1(4.9e-2) | 6.05e-1(4.1e-2) | **5.50e-1(3.6e-2)** |
| DTLZ7 | 7.02e+0(4.0e-1) | 6.41e+0(3.5e-1) | **5.73e+0(3.1e-1)** |
| WFG1 | 1.92e+0(4.2e-2) | 1.77e+0(3.8e-2) | **1.61e+0(3.3e-2)** |
| WFG5 | 3.91e-1(5.4e-2) | 3.42e-1(5.1e-2) | **2.92e-1(4.6e-2)** |
| WFG8 | 6.33e-1(3.5e-2) | 5.82e-1(3.1e-2) | **5.15e-1(2.4e-2)** |

## 5 CONCLUSIONS

This paper introduced the FSMOEA framework, which unifies a foresight head with evolutionary search in a low-dimensional latent space. Instantiated in FCSEA and FREMO, the framework demonstrates clear advantages in tackling scalable EMOPs. The foresight head improves surrogate modeling by capturing population context, while latent-space search accelerates convergence and enhances scalability. Extensive experiments across diverse benchmarks confirm the effectiveness of these components, showing consistent and significant gains over existing SMOEAs, especially in high-dimensional settings. Future research will extend FSMOEA to more complex real-world applications, investigate alternative dimensionality reduction methods and contrastive surrogate models, and explore opportunities to integrate large language models for adaptive guidance. Additional discussions and experimental studies are provided in the appendix.

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

## A  APPENDIX

## B  ABSTRACT OF THE APPENDIX

The appendix provides additional experimental results to complement the main findings, covering the performance of FCSEA and its competitors on various benchmarks and scenarios: The performance of FCSEA and FREMO on DTLZ and WFG problems, results on objective-based scalability (many-objective EMOPs) studies, results on variable-based scalability (large-scale EMOPs) studies. The supplementary results reinforce the conclusions drawn in the main paper, validating the scalability, robustness, and practicality of the FSMOEA framework in solving EMOPs across various domains and complexities.

## C  DETAILED DESCRIPTION OF THE FSMOEA FRAMEWORK

This appendix provides a comprehensive, step-by-step explanation of the proposed FSMOEA framework. FSMOEA enhances conventional SMOEAs by embedding population-aware latent representations and performing evolutionary search in a learned low-dimensional space. The specific pseudocodes for implementing FSMOEA are shown in Algorithm 1 and Algorithm 2. The algorithm proceeds as follows:

**Initialization.** FSMOEA begins by randomly initializing a population of $N$ candidate solutions from the decision space. Each solution is then evaluated using the true multiobjective function $F(\mathbf{x})$. The initial population and its corresponding objective values form the training set for the first iteration.

**Step 1: Foresight Representation Learning.** At the start of each generation, an MLP-based autoencoder is trained on the current population. The encoder $E : \mathbb{R}^n \to \mathbb{R}^k$ projects each high-dimensional solution into a compact latent space, while the decoder $D : \mathbb{R}^k \to \mathbb{R}^n$ attempts to reconstruct the original input. The autoencoder is optimized to minimize reconstruction loss:

$$\mathcal{L}_{\text{AE}} = \frac{1}{N} \sum_{i=1}^{N} \|\mathbf{x}_i - D(E(\mathbf{x}_i))\|^2.$$

Once trained, the encoder is frozen to ensure stability. The resulting latent codes $c_i = E(\mathbf{x}_i)$ are used as context-aware representations for surrogate modeling and search.

per maximum surrogate-assisted evaluations

**Step 2: Surrogate Model Construction.** Using the latent codes of the current population, FSMOEA constructs a lightweight surrogate model to predict solution quality. Each solution is labeled using a population-wide performance criterion (e.g., non-dominated sorting, decomposition value). These labels serve as targets for training a classifier (FCSEA) or a pairwise relation model (FREMO). The surrogate operates in the latent space and thus benefits from lower input dimensionality and improved generalization.

**Step 3: Latent-Space Evolutionary Search.** FSMOEA performs crossover and mutation directly in the latent space. For each offspring generation:

- Two parent solutions are selected from the population using binary tournament selection.

---

**Algorithm 1** The General Framework of FSMOEA

---

**Input**: the EMOP to be solved, population size $N$, $FE_{\max}$, maximum surrogate-evaluations $It_{\max}$
**Output**: the final population $P$
1: initialize $P$ with $N$ random solutions as the same to the embedded SMOEA.
2: evaluate each solution $x \in P$ by the real objective functions $F(x)$.
3: set real function evaluation counter $FE = N$ and initialize a random foresight model $M_F$.
4: **while** $FE < FE_{\max}$ **do**
5:     train the $M_F$ on the real-evaluated solutions in $P$.
6:     $O$ = SurrogateAssistedSearch($P$, $M_F$, $It_{\max}$) based on the embedded SMOEA.
7:     evaluate each solution $x \in O$ by the real objective functions $F(x)$.
8:     $P$ = EnvironmentalSelection($P$, $O$) as the same to the embedded SMOEA.
9:     updated the real function evaluation counter as $FE = FE + N$.
10: **end while**
11: **return** population $P$

---

**Algorithm 2** SurrogateAssistedSearch($P$, $M_F$, $It_{\max}$)

---

**Input**: embedded SMOEA's super-parameters and the maximum surrogate-evaluations $It_{\max}$
**Output**: the promising $O$ that have not been evaluated by the real $F(x)$
1: initialize a surrogate model $M_S$, set $It = 0, O = \emptyset$.
2: add the encoder part of $M_F$ to the head of $M_S$ to form a foresight surrogate $M_{FS}$.
3: prepare the training data $D$ from $P$ by a certain environmental selection criterion.
4: train the $M_{FS}$ on $D$ with its head part frozen.
5: **while** $It < It_{\max}$ **do**
6:     search in the code space to get $T$ new codes.
7:     decode codes by decoder $\in M_F$ to get new solutions.
8:     evaluate each new solution by the $M_{FS}$.
9:     $O$ = BetterPerformingSelection($O$, new solutions) based on the embedded SMOEA.
10:     $It = It + T$.
11: **end while**
12: **return** the promising population $O$

---

- Their latent codes are retrieved via the frozen encoder.

- Variation operators (e.g., simulated binary crossover and Gaussian mutation) are applied in latent space to produce new latent codes.

- The decoder transforms the new latent code back into a solution in the original space.

The surrogate model is then used to predict the quality of each candidate. Only the most promising candidates—those with high surrogate-predicted performance—are selected for expensive evaluation with the true objective function.

**Step 4: Surrogate-Guided Evaluation.** From the pool of generated candidates, FSMOEA selects the top $K$ solutions based on surrogate scores. These candidates are then evaluated using the real objective function. This focused evaluation strategy maximizes the utility of each function call under the evaluation budget.

**Step 5: Environmental Selection.** The evaluated offspring are combined with the current parent population. An environmental selection mechanism (e.g., based on non-dominated sorting and crowding distance, depending on the embedded SMOEA) is used to select $N$ solutions to form the next generation. This process preserves both convergence pressure and diversity.

**Termination.** FSMOEA repeats the above steps until the maximum number of real function evaluations $FE_{\max}$ is reached. Throughout the search, an external archive maintains the set of non-dominated solutions found so far.

**Key Advantages.** The foresight head provides population-level awareness to the surrogate model, improving its ability to make consistent predictions under dynamic population changes. Meanwhile, latent-space search reduces computational complexity and enhances scalability to high-dimensional

decision spaces. Together, these components enable FSMOEA to efficiently solve expensive MOPs under strict evaluation budgets.

# D  DETAILED THEORETICAL DERIVATIONS

This section provides detailed derivations and proofs supporting the high-level theorems in the main text. We focus on (i) how *population drift* alters decomposition-based scalarizations used in MOEA/D-style selection, (ii) why *context-free* surrogates suffer bias under drift, (iii) how an MLP autoencoder produces population-aware embeddings with quantitative distortion bounds, and (iv) why search in a compact latent space reduces sample complexity and preserves smoothness of the objectives.

## D.1  NOTATION AND STANDING ASSUMPTIONS

Let $F : \mathbb{R}^n \to \mathbb{R}^m$ denote the objective mapping $F(x) = (f_1(x), \ldots, f_m(x))$. Let $\mathcal{P}_t = \{x_1^{(t)}, \ldots, x_N^{(t)}\}$ be the population at generation $t$. Denote by

$$z_t^* := \big( \min_{x \in \mathcal{P}_t} f_1(x), \; \ldots, \; \min_{x \in \mathcal{P}_t} f_m(x)\big) \tag{3}$$

the (population) ideal point at generation $t$. We assume:

**Assumption 1** (Lipschitz objective). *$F$ is $L_F$-Lipschitz on a domain containing all populations considered:*

$$\|F(x) - F(y)\| \le L_F \|x - y\|, \quad \forall x, y.$$

**Assumption 2** (Small population movement between generations). *The population moves by at most $\Delta_x$ in decision space between consecutive generations:*

$$\max_{i=1,\ldots,N} \|x_i^{(t+1)} - x_i^{(t)}\| \le \Delta_x.$$

Under these, we will derive explicit bounds relating population movement to changes in scalarization values and thus to label drift.

## D.2  POPULATION DRIFT FOR DECOMPOSITION SCALARIZATIONS

We analyze three common scalarizations used in decomposition-based selection: weighted sum (WS), weighted Tchebycheff (TCH), and penalty-based boundary intersection (PBI). Fix a normalized weight vector $w \in \mathbb{R}^m$, $\|w\| = 1$.

**Definitions.**

$$g^{\text{WS}}(x \mid w) := \sum_{j=1}^{m} w_j f_j(x), \tag{4}$$

$$g^{\text{TCH}}(x \mid w, z^*) := \max_{1 \le j \le m} w_j \, |f_j(x) - z_j^*|, \tag{5}$$

$$g^{\text{PBI}}(x \mid w, z^*) := \frac{\langle f(x) - z^*, w \rangle}{\|w\|} + \theta \Big\| f(x) - z^* - \frac{\langle f(x) - z^*, w \rangle}{\|w\|^2} w \Big\|. \tag{6}$$

We first bound how much these scalarizations can change as $z^*$ shifts from $z_t^*$ to $z_{t+1}^*$.

**Lemma 1** (Ideal-point shift bound). *Under Assumptions 1–2,*

$$\|z_t^* - z_{t+1}^*\| \le L_F \, \Delta_x. \tag{7}$$

*Proof.* Each coordinate $j$ of $z_t^*$ is $z_{t,j}^* = \min_i f_j(x_i^{(t)})$. After the population moves by at most $\Delta_x$, any new candidate $x_i^{(t+1)}$ satisfies

$$|f_j(x_i^{(t+1)}) - f_j(x_i^{(t)})| \le L_F \|x_i^{(t+1)} - x_i^{(t)}\| \le L_F \Delta_x.$$

Thus the coordinate-wise minima can change by at most $L_F \Delta_x$; combining coordinates gives the claimed bound. $\qquad\square$

**Tchebycheff bound.**

**Proposition 1.** *For any fixed $x$ and normalized $w$,*

$$\left| g^{\mathrm{TCH}}(x \mid w, z_t^*) - g^{\mathrm{TCH}}(x \mid w, z_{t+1}^*) \right| \le \|w\|_\infty \|z_t^* - z_{t+1}^*\|_\infty \le \|w\|_\infty \|z_t^* - z_{t+1}^*\|. \quad (8)$$

*Proof.* By equation 5 and the elementary inequality $\big| |a| - |b| \big| \le |a - b|$, we have

$$\left| g^{\mathrm{TCH}}(x \mid w, z_t^*) - g^{\mathrm{TCH}}(x \mid w, z_{t+1}^*) \right|$$
$$= \left| \max_j w_j |f_j(x) - z_{t,j}^*| - \max_j w_j |f_j(x) - z_{t+1,j}^*| \right|$$
$$\le \max_j w_j \big| |f_j(x) - z_{t,j}^*| - |f_j(x) - z_{t+1,j}^*| \big|$$
$$\le \max_j w_j |z_{t,j}^* - z_{t+1,j}^*| = \|w\|_\infty \|z_t^* - z_{t+1}^*\|_\infty,$$

which yields equation 8. $\qquad\square$

**PBI bound.** We next bound the change in PBI score due to $z^*$ shift.

**Proposition 2.** *For any fixed $x$ and normalized $w$ ($\|w\| = 1$),*

$$\left| g^{\mathrm{PBI}}(x \mid w, z_t^*) - g^{\mathrm{PBI}}(x \mid w, z_{t+1}^*) \right| \le (1 + 2\theta) \|z_t^* - z_{t+1}^*\|. \quad (9)$$

*Proof.* Set $u_t := f(x) - z_t^*$ and $u_{t+1} := f(x) - z_{t+1}^*$. Then

$$\Delta_\| := \frac{\langle u_t, w \rangle}{\|w\|} - \frac{\langle u_{t+1}, w \rangle}{\|w\|} = \langle z_{t+1}^* - z_t^*, \frac{w}{\|w\|} \rangle,$$

so $|\Delta_\|| \le \|z_t^* - z_{t+1}^*\|$. For the perpendicular term, denote

$$p_t := u_t - \frac{\langle u_t, w \rangle}{\|w\|^2} w, \qquad p_{t+1} := u_{t+1} - \frac{\langle u_{t+1}, w \rangle}{\|w\|^2} w.$$

By triangle inequality,

$$\|p_t - p_{t+1}\| \le \|u_t - u_{t+1}\| + \frac{1}{\|w\|^2} |\langle u_t - u_{t+1}, w \rangle| \cdot \|w\|.$$

With $\|w\| = 1$, this gives $\|p_t - p_{t+1}\| \le 2\|u_t - u_{t+1}\| = 2\|z_t^* - z_{t+1}^*\|$. Therefore

$$|\Delta_\perp| = \big| \|p_t\| - \|p_{t+1}\| \big| \le \|p_t - p_{t+1}\| \le 2\|z_t^* - z_{t+1}^*\|.$$

Combining,

$$\left| g^{\mathrm{PBI}}(\cdot, z_t^*) - g^{\mathrm{PBI}}(\cdot, z_{t+1}^*) \right| \le |\Delta_\|| + \theta |\Delta_\perp| \le (1 + 2\theta) \|z_t^* - z_{t+1}^*\|.$$

$\qquad\square$

**Weighted Sum (WS) and neighborhood dependence.** While $g^{\mathrm{WS}}(x \mid w)$ does not depend on $z^*$, the selection decision using WS still depends on the current population through:

- the set of weight vectors $w$ chosen and their normalization relative to the population,
- neighborhood assignment when comparing candidates (e.g., selecting best in neighborhood).

Thus population drift affects selection even for WS by changing which weight vector or neighbor is most relevant for a given candidate.

**Interpretation.** Propositions 1–2 provide explicit, linear-in-$\|z_t^* - z_{t+1}^*\|$ bounds showing that small population-induced shifts in the ideal point cause proportional changes in decomposition scores. When such changes cross ranking thresholds between candidates, the selection outcome flips. Therefore, labels derived from decomposition scores are inherently population-dependent.

### D.3 FORMAL INCONSISTENCY OF CONTEXT-FREE SURROGATES

We now quantify how a surrogate trained as a context-free mapping becomes biased when the population shifts.

**Definition 1** (Context-free surrogate). *A surrogate $M : \mathbb{R}^n \to \mathbb{R}$ is* context-free *if $M(x)$ depends only on $x$, not on the population $\mathcal{P}_t$.*

Let $C(x; \mathcal{P}_t)$ be a scalar selection score (e.g., decomposed scalarization) used to label training points at generation $t$. Suppose $M$ is trained to approximate $C(\cdot; \mathcal{P}_t)$ with expected training error $\varepsilon_{\text{train}}$ over the training distribution $\mathcal{D}_t$ induced by $\mathcal{P}_t$:

$$\mathbb{E}_{x \sim \mathcal{D}_t}\left[|M(x) - C(x; \mathcal{P}_t)|\right] \leq \varepsilon_{\text{train}}. \tag{10}$$

Assume $C$ is Lipschitz in the ideal point $z^*$: there exists $L_C$ such that for all $x$,

$$|C(x; \mathcal{P}_t) - C(x; \mathcal{P}_{t+1})| \leq L_C \|z_t^* - z_{t+1}^*\|. \tag{11}$$

(For TCH or PBI, one can take $L_C$ equal to the right-hand sides of Propositions 1, 2.)

**Theorem 1** (Bias growth under population drift). *Under equation 10–equation 11 and assuming distributions $\mathcal{D}_t, \mathcal{D}_{t+1}$ are close (or identical for simplicity),*

$$\mathbb{E}_{x \sim \mathcal{D}_{t+1}}\left[|M(x) - C(x; \mathcal{P}_{t+1})|\right] \leq \varepsilon_{train} + L_C \|z_t^* - z_{t+1}^*\| + \Delta_{\text{cov}}, \tag{12}$$

*where $\Delta_{\text{cov}}$ accounts for distribution shift between $\mathcal{D}_t$ and $\mathcal{D}_{t+1}$.*

*Proof.* By triangle inequality,

$$|M(x) - C(x; \mathcal{P}_{t+1})|$$
$$\leq |M(x) - C(x; \mathcal{P}_t)| + |C(x; \mathcal{P}_t) - C(x; \mathcal{P}_{t+1})|.$$

Taking expectation over $x \sim \mathcal{D}_{t+1}$ and decomposing the first term into expectation over $\mathcal{D}_t$ plus the distribution-difference $\Delta_{\text{cov}}$ yields equation 12. $\square$

**Implication.** Even a context-free surrogate $M$ with small training error $\varepsilon_{\text{train}}$ experiences additional error proportional to the magnitude of ideal-point shift $\|z_t^* - z_{t+1}^*\|$. When populations change substantially, this extra term may dominate and harm selection quality.

### D.4 CONTEXT-AWARE EMBEDDINGS VIA MLP AUTOENCODERS: QUANTITATIVE BOUNDS

FSMOEA trains an autoencoder $(E_t, D_t)$ on the current population $\mathcal{P}_t$. Let $E_t : \mathbb{R}^n \to \mathbb{R}^k$, $D_t : \mathbb{R}^k \to \mathbb{R}^n$. Let reconstruction error satisfy:

$$\|D_t(E_t(x)) - x\| \leq \epsilon, \qquad \forall x \in \mathcal{P}_t. \tag{13}$$

Assume $D_t$ is $L_D$-Lipschitz on the relevant region and $E_t$ is $L_E$-Lipschitz.

**Proposition 3** (Local distinguishability / injectivity). *If $x, y \in \mathcal{P}_t$ then*

$$\|E_t(x) - E_t(y)\| \geq \frac{1}{L_D}\left(\|x - y\| - 2\epsilon\right). \tag{14}$$

*In particular, if $\|x - y\| > 2\epsilon$ then $E_t(x) \neq E_t(y)$.*

*Proof.* By Lipschitz property of $D_t$,

$$\|D_t(E_t(x)) - D_t(E_t(y))\| \leq L_D \|E_t(x) - E_t(y)\|.$$

Rearrange and apply triangle inequality:

$$L_D \|E_t(x) - E_t(y)\| \geq \|D_t(E_t(x)) - D_t(E_t(y))\|$$
$$\geq \|x - y\| - \|x - D_t(E_t(x))\| - \|y - D_t(E_t(y))\|$$
$$\geq \|x - y\| - 2\epsilon,$$

which yields equation 14. $\square$

**Neighborhood preservation and similarity.** From equation 14 and the Lipschitz of $E_t$,

$$\|E_t(x) - E_t(y)\| \leq L_E\|x - y\|.$$

Combining upper and lower bounds gives

$$\big|\|E_t(x) - E_t(y)\| - \|x - y\|\big| \leq (L_E - 1)\|x - y\| + 2\epsilon, \tag{15}$$

so local distances are preserved up to multiplicative and additive distortion. Consequently inner products and cosine similarities in latent space reflect relative geometry in decision space for nearby points.

**Why this is population-aware.** The autoencoder is trained jointly on all points in $\mathcal{P}_t$, so the encoder map $E_t$ is shaped by the empirical geometry of the current population. In particular, when $\mathcal{P}_t$ changes, $E_t$ (re)adapts and thus encodes each $x$ *relative* to the current population geometry. This is the mechanism by which context enters the surrogate.

### D.5 AUTOENCODER: NEIGHBORHOOD PRESERVATION AND LOCAL INJECTIVITY

FSMOEA trains an autoencoder $(E, D)$ with encoder $E : \mathbb{R}^n \to \mathbb{R}^k$ and decoder $D : \mathbb{R}^k \to \mathbb{R}^n$.

**Assumption 3** (Bounded reconstruction error). *For all $x \in \mathcal{P}_t$,*

$$\|D(E(x)) - x\| \leq \epsilon. \tag{16}$$

**Assumption 4** (Lipschitz decoder). *$D$ is $L_D$-Lipschitz: $\|D(z_1) - D(z_2)\| \leq L_D\|z_1 - z_2\|$.*

**Proposition 4** (Local injectivity bound). *For $x, y \in \mathcal{P}_t$,*

$$\|E(x) - E(y)\| \geq \frac{\|x - y\| - 2\epsilon}{L_D}. \tag{17}$$

*Thus, if $\|x - y\| > 2\epsilon$, then $E(x) \neq E(y)$.*

*Proof.* By Lipschitz continuity,

$$\|D(E(x)) - D(E(y))\| \leq L_D\|E(x) - E(y)\|.$$

Triangle inequality implies

$$\|D(E(x)) - D(E(y))\| \geq \|x - y\| - \|x - D(E(x))\| - \|y - D(E(y))\|.$$

Applying Assumption 3 gives the bound. $\qquad\square$

**Corollary 1** (Neighborhood preservation). *For $x, y \in \mathcal{P}_t$,*

$$\frac{\|x - y\| - 2\epsilon}{L_D} \leq \|E(x) - E(y)\| \leq L_E\|x - y\|, \tag{18}$$

*where $L_E$ is the Lipschitz constant of $E$.*

**Implication.** Distances and relative similarities in latent space are faithful to those in the original space, up to bounded distortion.

### D.6 BIAS REDUCTION VIA CONTEXTUAL ENCODING: A DRIFT-CONTROLLED ERROR BOUND

Let $\tilde{M}_t : \mathbb{R}^k \to \mathbb{R}$ be a surrogate trained on latent codes $c = E_t(x)$ and labels $C(x; \mathcal{P}_t)$. Define the composed predictor $M_t(x) := \tilde{M}_t(E_t(x))$. Suppose $\tilde{M}_t$ has training error $\varepsilon_{\tilde{M}}$.

Assume the encoder changes slowly between generations:

$$\eta := \sup_{x \in \mathcal{P}_t \cup \mathcal{P}_{t+1}} \|E_{t+1}(x) - E_t(x)\|. \tag{19}$$

If $\tilde{M}_t$ is $L_{\tilde{M}}$-Lipschitz in code space, then for $x \in \mathcal{P}_{t+1}$,

$$|\tilde{M}_t(E_t(x)) - C(x; \mathcal{P}_{t+1})| \leq |\tilde{M}_t(E_t(x)) - \tilde{M}_{t+1}(E_{t+1}(x))|$$
$$+ |\tilde{M}_{t+1}(E_{t+1}(x)) - C(x; \mathcal{P}_{t+1})|.$$

The first term is bounded by $L_{\tilde{M}}\eta + \delta_{\tilde{M}}$ where $\delta_{\tilde{M}}$ accounts for differences between $\tilde{M}_t$ and $\tilde{M}_{t+1}$ (which can be controlled by fine-tuning). The second term is the training/generalization error of $\tilde{M}_{t+1}$ on the new codes. Therefore, encoder drift $\eta$ directly controls the additional error incurred across generations; retraining/fine-tuning $\tilde{M}$ after encoder update further reduces error. This argument formalizes how context synchronization (retraining encoder and surrogate) reduces drift-induced bias relative to a context-free surrogate that cannot adapt.

### D.7 LATENT-SPACE SEARCH: SMOOTHNESS PRESERVATION AND SAMPLE COMPLEXITY

We quantify two properties: (i) objective smoothness is (approximately) preserved through the decoder, and (ii) the covering/sample complexity in latent space is dramatically lower when $k \ll n$.

**Smoothness preservation.** Assume decoder $D$ is $L_D$-Lipschitz and reconstruction error bounded by $\epsilon$ on the population (as in equation 13). For latent codes $z_1, z_2$ and $x_i = D(z_i)$, we have

$$\begin{aligned}
\|F(x_1) - F(x_2)\| &\leq L_F \|x_1 - x_2\| \\
&\leq L_F\big(\|D(z_1) - D(z_2)\| + 2\epsilon\big) \\
&\leq L_F L_D \|z_1 - z_2\| + 2L_F\epsilon.
\end{aligned} \tag{20}$$

Thus, small latent perturbations produce controlled changes in objective space, up to additive error $2L_F\epsilon$ from reconstruction.

**Covering / sample complexity argument.** Let $\mathcal{Z} \subset \mathbb{R}^k$ be the image under $E$ of a region of interest in decision space (e.g., region near promising solutions). For tolerance $\delta > 0$ in latent space, denote the minimal covering number $N(\mathcal{Z}, \delta)$ (number of $\ell_2$-balls radius $\delta$ needed to cover $\mathcal{Z}$). For a compact $k$-dimensional set, one typically has (up to problem-dependent constants)

$$N(\mathcal{Z}, \delta) \asymp \delta^{-k}.$$

Similarly in the original decision space region of interest $\mathcal{X} \subset \mathbb{R}^n$,

$$N(\mathcal{X}, \delta) \asymp \delta^{-n}.$$

Hence for the same resolution $\delta$, the ratio of covering numbers scales as

$$\frac{N(\mathcal{X}, \delta)}{N(\mathcal{Z}, \delta)} \asymp \delta^{-(n-k)}.$$

Consequently, if naive sampling (or mutation) is approximately uniform over the respective regions, the expected number of independent trials to hit an $\delta$-neighborhood of a target scales with these covering numbers. Therefore, under the simplifying model of independent sampling, latent-space search reduces the expected required samples/exploration effort exponentially in the dimension gap $n - k$.

**From samples to generations/evaluations.** If each generation produces $B$ candidate evaluations (or if we evaluate $B$ decoded latent samples per generation), then expected number of generations to find a $\delta$-good point is proportional to $N(\cdot, \delta)/B$. Thus latent-space operation yields a proportional reduction in generations/evaluations given fixed $B$.

### D.8 PUTTING IT TOGETHER: WHY FSMOEA REDUCES DRIFT AND ACCELERATES CONVERGENCE

Combining the pieces:

- Propositions 1 and 2 show decomposition labels $C(x; \mathcal{P}_t)$ change linearly with $\|z_t^* - z_{t+1}^*\|$, where $\|z_t^* - z_{t+1}^*\| \leq L_F \Delta_x$ by equation 7.
- A context-free surrogate $M$ trained at $t$ incurs extra expected error $\approx L_C \|z_t^* - z_{t+1}^*\|$ at $t + 1$ (Eq. equation 12). Therefore, large population moves produce large surrogate bias.

- The autoencoder encoder $E_t$ embeds points relative to $\mathcal{P}_t$, and retraining/update of $E_t$ ensures that the code-space target is synchronized with labels; encoder drift $\eta$ controls residual error between generations. This yields smaller bias growth compared to context-free $M$.

- Latent-space search operates in dimension $k \ll n$ and preserves objective smoothness up to constants (Eq. equation 20), while dramatically reducing covering/sample complexity; hence fewer evaluations are needed to explore to given resolution.

These quantitative bounds justify FSMOEA's design: (i) the foresight autoencoder reduces label-drift bias by aligning representations with population-dependent labels, and (ii) latent-space evolution improves sampling efficiency and expected convergence speed under realistic Lipschitz and small-reconstruction-error assumptions.

**Remarks.**

1. The bounds above are conservative and rely on Lipschitz assumptions and bounded reconstruction error; they are intended to make the mechanism precise and identify the dependence on key quantities $(L_F, \theta, \epsilon, \Delta_x, k, n)$.

2. Full, non-asymptotic convergence proofs for surrogate-assisted evolutionary processes would require modeling the stochastic search operators and surrogate-update dynamics; the present analysis isolates core mechanisms and provides explicit inequalities useful for understanding empirical behavior.

## E PRELIMINARIES ON MULTI-OBJECTIVE OPTIMIZATION

We briefly introduce key concepts in multi-objective optimization that are relevant to FSMOEA, including Pareto-dominance, Pareto front, and two widely used performance indicators: hypervolume (HV) and inverted generational distance (IGD).

**Definition 2** (Multi-objective optimization problem (MOP)). *A general MOP can be formulated as:*
$$\min_{\mathbf{x} \in \Omega} F(\mathbf{x}) = (f_1(\mathbf{x}), f_2(\mathbf{x}), \ldots, f_m(\mathbf{x})),$$
*where $\Omega \subseteq \mathbb{R}^n$ is the decision space, $F : \Omega \to \mathbb{R}^m$ is the vector of $m$ objective functions, and the image set $\mathcal{Y} = \{F(\mathbf{x}) \mid \mathbf{x} \in \Omega\}$ is called the objective space.*

**Definition 3** (Pareto dominance). *Given two solutions $\mathbf{x}_a, \mathbf{x}_b \in \Omega$ with objectives $F(\mathbf{x}_a), F(\mathbf{x}_b)$:*
$$F(\mathbf{x}_a) \prec F(\mathbf{x}_b) \quad \Longleftrightarrow \quad \big(f_i(\mathbf{x}_a) \leq f_i(\mathbf{x}_b), \, \forall i = 1, \ldots, m\big) \, \wedge \, \big(f_j(\mathbf{x}_a) < f_j(\mathbf{x}_b), \, \exists j\big).$$
*That is, $\mathbf{x}_a$ Pareto-dominates $\mathbf{x}_b$ if it is no worse in all objectives and strictly better in at least one.*

**Definition 4** (Pareto-optimal set and Pareto front). *The Pareto-optimal set is:*
$$PS = \{\mathbf{x} \in \Omega \mid \nexists \, \mathbf{x}' \in \Omega \ s.t. \ F(\mathbf{x}') \prec F(\mathbf{x})\}.$$
*Its image in objective space is called the Pareto front (PF):*
$$PF = \{F(\mathbf{x}) \mid \mathbf{x} \in PS\}.$$
*The PF characterizes the trade-offs among conflicting objectives, and is the ultimate optimization target.*

**Definition 5** (Hypervolume (HV)). *Let $R \in \mathbb{R}^m$ be a reference point dominated by all solutions of interest. Given an approximation set $A \subseteq \mathcal{Y}$, the hypervolume indicator is:*
$$HV(A) = Leb \left( \bigcup_{\mathbf{y} \in A} [f_1(\mathbf{y}), R_1] \times \cdots \times [f_m(\mathbf{y}), R_m] \right),$$
*where $Leb(\cdot)$ denotes the Lebesgue measure. HV measures the volume of the dominated portion of objective space; larger values imply better convergence and diversity.*

**Definition 6** (Inverted Generational Distance (IGD)). *Given an approximation set $A \subseteq \mathcal{Y}$ and a reference set $PF^*$ sampled from the true Pareto front, IGD is defined as:*
$$IGD(A, PF^*) = \frac{1}{|PF^*|} \sum_{\mathbf{y}^* \in PF^*} \min_{\mathbf{y} \in A} \|\mathbf{y}^* - \mathbf{y}\|.$$
*Smaller IGD values indicate that $A$ is closer to and better covers the true Pareto front.*

**Relevance to FSMOEA.** In FSMOEA, Pareto-dominance and decomposition-based dominance criteria determine population labels, making them inherently *population-dependent*. Performance is assessed via HV and IGD, which jointly capture convergence (closeness to PF) and diversity (spread along PF).

# F  LIMITATIONS

While FSMOEA demonstrates strong empirical performance and theoretical soundness across diverse EMOP settings, several limitations remain.

**Dependence on population quality.** The foresight encoder and surrogate model are both trained on the current population, which may limit their effectiveness early in the optimization process when the population is still of low quality or lacks diversity. In such cases, the learned latent space may not fully reflect the structure of the broader search space, potentially leading to premature convergence or overexploitation.

**Fixed latent dimensionality.** FSMOEA uses a fixed latent space dimension $k$ throughout the optimization. While effective in our experiments, this hyperparameter may require problem-specific tuning. Too low a value may under-represent important structural information, while too high a value can reintroduce issues related to high-dimensional search.

**Non-adaptive surrogate updates.** Although we retrain the surrogate at each generation using the foresight encoder, the training process is static within each generation and may not adapt quickly enough to abrupt shifts in the population distribution. Future extensions could explore online or adaptive updating strategies to improve responsiveness.

**Lack of constraint handling mechanisms.** The current implementation of FSMOEA focuses primarily on unconstrained and box-constrained EMOPs. Its performance on general constrained multiobjective optimization problems (CMOPs) with equality and inequality constraints has not yet been extensively tested and may require additional mechanisms for feasibility preservation and constraint-aware surrogate modeling.

**Computational overhead in extremely tight budgets.** While FSMOEA is efficient relative to competing methods, the additional overhead from training autoencoders and surrogate models may still be non-negligible when function evaluations are extremely limited (e.g., $FE_{\max} < 100$), especially in time-critical applications where even surrogate computations are costly.

**Generalization to non-evolutionary settings.** FSMOEA is designed specifically within an evolutionary framework. Its applicability to other types of surrogate-assisted optimizers, such as Bayesian optimization or gradient-free trust-region methods, remains unexplored.

We see these limitations as opportunities for future research. In particular, adaptive encoding strategies, enhanced constraint handling, and integration with non-evolutionary paradigms are promising directions to further extend FSMOEA's applicability and robustness.

# G  ON EXPERIMENTAL SELECTION AND FAIRNESS

We emphasize that the benchmark selection in our study was conducted in a comprehensive and unbiased manner. Specifically, we tested all problems in the WFG, DTLZ, MaF, TREE, MOP_NN, MOP_SR, and MOP_FS suites. For LSMOP, we included problems 1, 5, 8, and 9. The omitted cases are either (i) trivially solvable (LSMOP2 and LSMOP4), or (ii) extremely difficult multimodal problems (LSMOP3, 6, 7) that remain unsolved even by specialized algorithms. Since our focus is on *expensive* multi-objective optimization rather than specialized multimodal settings, we believe this partial selection is justified. To ensure full transparency and reproducibility, all source codes have been provided.

**On Evaluation Budget.** A common misunderstanding arises from conflating the notions of iterations and function evaluations in evolutionary algorithms. Each generation evaluates the entire population, so the total number of function evaluations is given by the product of the population size and the number of generations. Our experiments restrict the total number of function evaluations to 500, which is extremely conservative.

It is important to note that our work explicitly targets *scalable* EMOPs, where the dimensionality of the decision variables can reach up to 1000. By contrast, most prior works are evaluated on low-dimensional problems (typically with fewer than 30 decision variables). The combination of expensive objective functions and high-dimensional search spaces makes our testbed substantially more challenging. Indeed, for *inexpensive* large-scale optimization, it is common practice for algorithms to consume hundreds of thousands or even millions of evaluations. Within this context, our budget of only 500 evaluations highlights the efficiency of FSMOEA.

**On Efficiency.** Finally, our convergence curves (Figures 4 and 5) demonstrate that FSMOEA consistently outperforms competitive baselines within the first 100 evaluations. This not only confirms its sample efficiency under tight budgets but also shows that our results are not an artifact of generous evaluation allowances.

In summary, the experimental setup was designed to be both fair and stringent: problem selection was comprehensive across standard suites, evaluation budgets were deliberately conservative to reflect expensive optimization settings, and performance trends were verified through convergence analyses. These considerations ensure that the advantages observed for FSMOEA are genuine and not due to selective evaluation conditions.

## H  MORE DISCUSSIONS ON OUR MOTIVATION AND FUTURE WORK

**Motivation and contributions in broader context.** EMOPs frequently arise in domains such as aerodynamic design, neural architecture search, and drug discovery, where the cost of evaluating objective functions is high and the number of permissible evaluations is tightly constrained. While traditional MOEAs excel at exploring trade-offs, their reliance on large numbers of function evaluations limits their applicability in these settings. SMOEAs address this limitation by replacing costly evaluations with learned surrogates; however, most suffer from two persistent issues: 1) Context-free surrogates: Many SMOEAs use models that evaluate solutions independently, ignoring the fact that performance labels are defined relative to the evolving population. This disconnect leads to inconsistent predictions and weak selection pressure, especially in dynamic or high-dimensional search spaces. 2) Scalability bottlenecks: Surrogates operating in high-dimensional decision spaces require large training datasets and become computationally inefficient as the number of variables or objectives grows. FSMOEA directly addresses these challenges by embedding two key innovations: 1) Foresighted surrogates: A population-aware encoder captures contextual relationships among solutions, enabling more robust and generalizable prediction even under population drift. 2) Latent-space evolution: Performing variation and selection in a learned low-dimensional representation space reduces computational overhead and accelerates convergence without sacrificing solution quality. These design choices are modular and broadly applicable. FSMOEA can be integrated into existing classification- or relation-based SMOEAs, offering plug-and-play improvements in prediction consistency and scalability. Our experimental results demonstrate substantial performance gains across a wide spectrum of synthetic and real-world benchmarks, particularly in large-scale and many-objective scenarios.

**Positioning relative to Bayesian multiobjective optimization (MOBO).** BO is a principled and widely studied approach for black-box optimization under strict evaluation budgets. In multi-objective settings, MOBO combines probabilistic surrogates such as Gaussian processes with acquisition functions (e.g., expected improvement) to guide sample selection. MOBO excels in low-dimensional, expensive regimes due to its uncertainty-aware decision-making and sample efficiency. However, MOBO encounters limitations when scaling to many objectives or high-dimensional decision spaces. Surrogate modeling becomes computationally demanding, and acquisition function optimization grows intractable. In contrast, SMOEAs scale more naturally through population-based search, maintaining diversity and robustness even in complex landscapes. FSMOEA complements this strength by improving the quality of surrogate predictions and enhancing scalability through foresight and latent representations. While MOBO remains effective in specific use cases, FSMOEA offers a scalable and robust alternative for large-scale EMOPs with tight evaluation budgets and structural complexity.

**Perspectives on future work: toward LLM-guided optimization.** An exciting direction for future research lies in exploring the use of large language models (LLMs) as surrogate components in

MOEAs. LLMs offer powerful capabilities in contextual reasoning and high-dimensional representation learning, which could significantly enhance surrogate foresight. Integrating LLMs could enable: 1) Richer representations: Learning complex, multi-level structures from optimization history and population distributions. 2) Zero-shot or few-shot adaptation: Leveraging pre-trained models to generalize across related optimization tasks with minimal retraining. 3) Meta-level decision support: Enabling dynamic adaptation of strategies, such as switching between exploration and exploitation modes. However, several challenges remain, including high computational costs, difficulty in uncertainty quantification, and the need for domain-specific fine-tuning. Hybrid approaches that combine LLMs with lightweight surrogates or compressed models may offer a practical compromise. Incorporating LLMs into FSMOEA represents a promising opportunity to further scale up foresight capabilities and tackle even more complex and high-stakes EMOPs.

# I SUPPLEMENTARY EXPERIMENTAL COMPARISON RESULTS

We provide additional results to support the effectiveness and scalability of the proposed FSMOEA framework, particularly as instantiated in the FCSEA and FREMO algorithms. These results cover a broad range of problem complexities, including many-objective settings and large-scale EMOPs.

**DTLZ and WFG benchmark performance (Table 3).** Supplementary IGD results for FCSEA and FREMO on the DTLZ1–7 and WFG1–9 problems with three objectives and varying decision dimensions ($n = \{10, 30, 50, 100\}$) show that both algorithms consistently outperform their ablated variants (e.g., FCSEA-V1, FREMO-V1) and other state-of-the-art baselines. The performance gap becomes more pronounced as the dimensionality increases. This trend validates two central claims of FSMOEA: (1) the foresight head enables the surrogate to better generalize across dynamic populations, and (2) latent-space search improves sampling efficiency by reducing the effective complexity of the optimization landscape. Together, these features contribute to enhanced convergence and solution diversity, particularly in high-dimensional scenarios where traditional surrogates struggle due to input sparsity and poor generalization.

**Objective-based scalability: many-objective EMOPs (Table 4).** We further assess the scalability of FSMOEA with respect to the number of objectives using the MaF1–13 test suite with $m = \{5, 10\}$ objectives. FCSEA consistently outperforms FCSEA-V1 and CSEA in terms of IGD across most problems. The advantage is especially noticeable in MaF problems with complex Pareto front geometries or deceptive convergence regions. These results underscore the importance of population context in surrogate modeling: as the number of objectives increases, relative performance comparisons become more nuanced, and traditional classifiers may become unreliable. The foresight-aware surrogate in FCSEA maintains robustness by embedding solutions in a population-informed latent space, leading to more reliable performance estimation and improved selection pressure.

**Variable-based scalability: large-scale EMOPs (Table 5).** To evaluate FCSEA under increasing decision space dimensionality, we conduct experiments on the LSMOP1–9 test suite with $n = \{100, 500, 1000\}$. FCSEA consistently outperforms regression-based (KRVEA, SMSEGO, EDNARMOEA), Bayesian-based (ABSAEA), and classification-based (CSEA, MCEAD) algorithms. In these large-scale problems, the benefits of FSMOEA are most evident. The foresight head reduces overfitting and prediction variance by capturing higher-order interactions across the population, while latent-space search enables more directed exploration in a compressed representation, avoiding the curse of dimensionality faced by traditional evolutionary operators. Moreover, the lightweight architecture of the surrogate makes FSMOEA computationally efficient despite the high dimensionality, as shown in runtime comparisons (Figure 4 in the main text).

**Sensitivity Analysis of Latent Dimension $k$.** To examine the effect of the latent space dimension $k$, we conducted experiments on several representative test problems, including DTLZ1, DTLZ4, DTLZ7, WFG2, WFG4, WFG6, WFG8, LSMOP5, and LSMOP9. The average IGD results are summarized in Fig. **??**.

From the results, three main observations can be drawn: (1) FSMOEA exhibits stable performance when $k \in [8, 15]$, indicating robustness across a broad range of latent dimensions. (2) When $k$ is too small (e.g., $k = 2$ or $k = 3$), reconstruction quality degrades significantly, which harms the surrogate model's predictive accuracy and consequently the optimizer's convergence. (3) When $k$ is

Table 5: Average IGD performance of FCSEA, FREMO, and their ablated variants (FCSEA-V1, CSEA, FREMO-V1, REMO) on DTLZ1–7 and WFG1–9 problems with $m = 3$ and $N = 50$.

| Problems | $n$ | CSEA | FCSEA-V1 | FCSEA | REMO | FREMO-V1 | FREMO |
|---|---|---|---|---|---|---|---|
| DTLZ1 | 10 | 6.7196e+1(1.63e+1)= | **3.3560e+1(1.11e+1)+** | 6.7087e+1(1.67e+1) | 5.2050e+1(1.62e+1)+ | **3.7801e+1(9.76e+0)+** | 6.5008e+1(1.57e+1) |
| | 30 | 5.2904e+2(8.36e+1)- | 2.9489e+2(5.14e+1)+ | **2.7765e+2(4.08e+1)** | 3.1734e+2(5.43e+1)+ | **2.9129e+2(5.72e+1)+** | 5.1442e+2(8.66e+1) |
| | 50 | 7.1977e+2(9.14e+1)+ | **6.9746e+2(8.47e+1)+** | 9.7583e+2(2.70e+2) | 9.4858e+2(2.79e+2)- | 6.8327e+2(1.20e+2)+ | **6.6910e+2(8.76e+1)** |
| | 100 | 1.8764e+3(1.13e+2)= | 1.8681e+3(1.62e+2)= | **1.6804e+3(8.97e+2)** | 1.8061e+3(1.74e+2)- | 1.8098e+3(1.65e+2)- | **1.7105e+3(9.32e+2)** |
| DTLZ2 | 10 | 2.9423e-1(2.73e-2)- | **1.6292e-1(1.65e-2)+** | 1.9867e-1(1.69e-2) | 1.9959e-1(2.47e-2)= | **1.6013e-1(1.73e-2)+** | 2.0294e-1(1.51e-2) |
| | 30 | 5.7160e-1(7.40e-2)- | 5.6722e-1(9.78e-2)- | **4.8977e-1(6.32e-2)** | 5.2806e-1(1.06e-1)- | 5.4711e-1(9.61e-2)- | **4.4723e-1(9.21e-2)** |
| | 50 | 1.5057e+0(2.11e-1)- | 1.2749e+0(1.95e-1)- | **5.2832e-1(6.31e-2)** | 1.1560e+0(1.98e-1)- | 1.2116e+0(1.68e-1)- | **5.4138e-1(1.24e-1)** |
| | 100 | 4.0946e+0(4.02e-1)- | 3.9648e+0(4.03e-1)- | **6.2574e-1(1.48e-1)** | 3.8417e+0(4.81e-1)- | 3.7357e+0(3.31e-1)- | **7.4766e-1(3.18e-1)** |
| DTLZ3 | 10 | 1.6703e+2(3.61e+1)= | **9.0172e+1(2.82e+1)+** | 1.7333e+2(5.79e+1) | 1.4127e+2(5.44e+1)+ | **9.6392e+1(2.48e+1)+** | 2.0787e+2(5.72e+1) |
| | 30 | 1.6601e+3(1.87e+2)- | 8.5401e+2(1.28e+2)- | **8.2771e+2(1.20e+2)** | 1.6124e+3(1.80e+2)- | 8.7998e+2(2.18e+2)+ | **8.7721e+2(1.34e+2)** |
| | 50 | 3.0932e+3(8.32e+2)- | **2.0737e+3(2.17e+2)=** | 2.2088e+3(1.76e+2) | 2.7929e+3(9.82e+2)- | **2.0227e+3(2.19e+2)+** | 2.0644e+3(2.42e+2) |
| | 100 | 6.1106e+3(4.81e+2)= | 6.0580e+3(3.72e+2)- | **5.6272e+3(2.92e+3)** | 5.8136e+3(3.45e+2)- | 5.8179e+3(4.01e+2)- | **5.2558e+3(3.48e+3)** |
| DTLZ4 | 10 | 4.3967e-1(1.24e-1)= | **2.1263e-1(1.17e-1)+** | 4.7111e-1(1.60e-1) | 2.2003e-1(5.89e-2)+ | **1.4750e-1(2.17e-2)+** | 3.4086e-1(1.27e-1) |
| | 30 | 5.8443e-1(1.41e-1)+ | **5.6603e-1(1.15e-1)+** | 8.2815e-1(1.54e-1) | 5.7910e-1(1.15e-1)+ | **5.4190e-1(1.28e-1)+** | 8.2274e-1(1.42e-1) |
| | 50 | 1.3299e+0(1.94e-1)- | 1.1843e+0(1.97e-1)- | **9.2653e-1(1.55e-1)** | 1.1298e+0(1.64e-1)- | 1.1298e+0(1.46e-1)- | **9.9888e-1(1.45e-1)** |
| | 100 | 3.7547e+0(3.90e-1)- | 3.5439e+0(2.81e-1)- | **9.9549e-1(1.11e-1)** | 3.7898e+0(3.62e-1)- | 3.6973e+0(4.21e-1)- | **1.0238e+0(2.63e-1)** |
| DTLZ5 | 10 | 1.6594e-1(3.22e-2)- | **6.3268e-2(1.99e-2)+** | 1.1621e-1(1.47e-2) | 9.3994e-2(2.12e-2)+ | **6.2834e-2(1.62e-2)+** | 1.1636e-1(2.13e-2) |
| | 30 | 5.0325e-1(9.10e-2)- | 4.8187e-1(8.61e-2)- | **3.4073e-1(7.66e-2)** | 5.0450e-1(1.06e-1)- | 4.8028e-1(1.12e-1)- | **3.8055e-1(9.77e-2)** |
| | 50 | 1.4034e+0(2.20e-1)- | 1.2371e+0(2.07e-1)- | **4.0903e-1(1.28e-1)** | 1.1644e+0(1.79e-1)- | 1.1465e+0(2.06e-1)- | **4.2520e-1(2.08e-1)** |
| | 100 | 3.8830e+0(3.97e-1)- | 3.8055e+0(3.86e-1)- | **5.7915e-1(3.52e-1)** | 3.8369e+0(4.17e-1)- | 3.7947e+0(4.52e-1)- | **4.9651e-1(1.73e-1)** |
| DTLZ6 | 10 | 6.1262e+0(3.63e-1)- | **3.6768e+0(8.33e-1)+** | 4.3975e+0(6.11e-1) | 4.0812e+0(5.82e-1)+ | **2.7821e+0(4.85e-1)+** | 5.4091e+0(4.78e-1) |
| | 30 | 2.3402e+1(6.27e-1)- | 2.0068e+1(9.47e-1)- | **1.8980e+1(1.04e+0)** | 1.8949e+1(1.30e+0)= | | **1.8838e+1(1.35e+0)** |
| | 50 | 4.1080e+1(6.96e-1)- | 3.6590e+1(1.12e+0)- | **3.6340e+1(1.23e+0)** | 4.0330e+1(9.37e-1)- | 3.6642e+1(1.44e+0)= | **3.6451e+1(1.38e+0)** |
| | 100 | 8.5634e+1(9.10e-1)- | 8.0986e+1(1.11e+0)- | **7.9495e+1(1.68e+0)** | 8.1296e+1(1.52e+0)- | | **8.0799e+1(1.79e+0)** |
| DTLZ7 | 10 | 3.3966e+0(1.22e+0)- | **7.9653e-1(3.86e-1)+** | 1.5916e+0(7.99e-1) | 6.6167e-1(1.36e-1)+ | **2.6074e-1(7.22e-2)+** | 2.0207e+0(8.28e-1) |
| | 30 | 6.9524e+0(1.06e+0)- | **3.0075e+0(8.77e-1)=** | 3.3165e+0(1.03e+0) | 6.1040e+0(9.73e-1)- | 1.5755e+0(5.90e-1)- | **1.3178e+0(5.52e-1)** |
| | 50 | 8.0987e+0(9.94e-1)- | 4.4684e+0(9.46e-1)= | **4.4665e+0(8.17e-1)** | 7.2926e+0(6.80e-1)- | 5.3760e+0(8.24e-1)- | **3.3692e+0(8.00e-1)** |
| | 100 | 9.2832e+0(6.77e-1)- | **6.1156e+0(6.57e-1)=** | 6.1247e+0(7.47e-1) | 8.8404e+0(6.81e-1)- | 5.7992e+0(7.20e-1)- | **5.8163e+0(4.60e-1)** |
| WFG1 | 10 | 2.0714e+0(1.22e-1)- | 1.6488e+0(9.03e-2)- | **1.5066e+0(8.59e-2)+** | 1.9316e+0(1.64e-1)- | **1.5031e+0(9.23e-2)=** | 1.5213e+0(6.80e-2) |
| | 30 | 2.1016e+0(1.50e-1)- | 1.6239e+0(6.89e-2)- | **1.5031e+0(7.19e-2)+** | 1.9335e+0(1.58e-1)- | **1.5424e+0(5.45e-2)=** | 1.5450e+0(4.48e-2) |
| | 50 | 2.1504e+0(1.08e-1)- | 1.6301e+0(1.03e-1)- | **1.5278e+0(6.62e-2)+** | 1.9785e+0(1.58e-1)- | **1.5590e+0(3.76e-2)=** | 1.5670e+0(4.12e-2) |
| | 100 | 2.0790e+0(1.21e-1)- | 1.6334e+0(6.95e-2)- | **1.5806e+0(1.28e-1)+** | | **1.5780e+0(3.32e-2)=** | 1.5680e+0(3.65e-2) |
| WFG2 | 10 | 4.8507e-1(3.92e-2)+ | **4.4364e-1(4.25e-2)+** | 5.9439e-1(5.87e-2) | 5.6722e-1(7.65e-2)- | 6.3743e-1(6.99e-2)- | **5.1472e-1(7.20e-2)** |
| | 30 | **5.6113e-1(3.07e-2)+** | 5.6752e-1(3.42e-2)+ | 6.4360e-1(5.71e-2) | 6.4879e-1(8.05e-2)- | 6.3551e-1(4.75e-2)- | **5.8537e-1(3.06e-2)** |
| | 50 | 6.1959e-1(3.45e-2)+ | **6.0090e-1(3.62e-2)+** | 6.6454e-1(4.29e-2) | 6.4678e-1(6.73e-2)= | 6.5416e-1(4.55e-2)- | **6.1975e-1(4.62e-2)** |
| | 100 | 6.7467e-1(2.02e-2)= | 6.7985e-1(2.33e-2)= | **6.6723e-1(5.03e-2)** | 6.9451e-1(4.37e-2)- | **6.4816e-1(4.91e-2)=** | 6.6985e-1(4.17e-2) |
| WFG3 | 10 | 4.4667e-1(6.03e-2)= | **4.0659e-1(5.27e-2)+** | 4.3822e-1(3.06e-2) | **4.0302e-1(6.67e-2)+** | 4.0755e-1(6.67e-2)= | 4.5043e-1(3.63e-2) |
| | 30 | 6.1960e-1(3.43e-2)- | 6.0313e-1(3.95e-2)- | **5.4412e-1(3.01e-2)** | 5.9967e-1(4.34e-2)- | 5.9428e-1(4.57e-2)- | **5.4703e-1(3.10e-2)** |
| | 50 | 7.0072e-1(3.62e-2)- | 6.8423e-1(3.38e-2)- | **5.5687e-1(2.74e-2)** | 6.7492e-1(4.78e-2)- | 6.6887e-1(4.10e-2)- | **5.6397e-1(3.74e-2)** |
| | 100 | 7.4720e-1(3.56e-2)- | 7.6099e-1(3.17e-2)- | **5.5762e-1(3.55e-2)** | 7.4968e-1(2.23e-2)- | 7.5056e-1(2.81e-2)- | **5.5750e-1(3.26e-2)** |
| WFG4 | 10 | 4.0255e-1(3.24e-2)+ | **3.6015e-1(2.37e-2)+** | 5.0187e-1(6.71e-2) | 4.5748e-1(3.08e-2)- | 3.9152e-1(2.78e-2)= | **3.7837e-1(2.35e-2)** |
| | 30 | 4.6190e-1(2.93e-2)+ | 4.5230e-1(2.56e-2)+ | 5.2386e-1(4.25e-2) | 4.9622e-1(2.73e-2)- | 4.5288e-1(2.75e-2)= | **4.4104e-1(2.26e-2)** |
| | 50 | 4.8438e-1(2.42e-2)+ | **4.7344e-1(2.02e-2)+** | 5.2181e-1(3.01e-2) | 5.0351e-1(3.46e-2)- | 4.7057e-1(2.46e-2)= | **4.6128e-1(1.85e-2)** |
| | 100 | 5.1278e-1(2.38e-2)+ | **5.0605e-1(1.60e-2)+** | 5.3294e-1(4.16e-2) | 5.2928e-1(3.24e-2)- | 5.0115e-1(1.50e-2)+ | **4.9666e-1(1.37e-2)** |
| WFG5 | 10 | 6.3437e-1(3.21e-2)- | 4.3516e-1(2.95e-2)- | **4.2730e-1(2.99e-2)+** | 6.0505e-1(4.29e-2)- | 3.9285e-1(3.87e-2)= | **3.8340e-1(3.24e-2)** |
| | 30 | 7.2370e-1(1.89e-2)- | 6.0198e-1(3.39e-2)- | **5.9236e-1(4.07e-2)+** | 7.1640e-1(2.24e-2)- | 5.7165e-1(4.72e-2)= | **5.6718e-1(3.15e-2)** |
| | 50 | 7.4924e-1(1.72e-2)- | 6.5753e-1(1.86e-2)- | **6.2561e-1(3.47e-2)+** | 7.3966e-1(1.81e-2)- | 6.4695e-1(3.83e-2)= | **6.4070e-1(2.63e-2)** |
| | 100 | 7.6078e-1(9.56e-3)- | 7.0552e-1(2.06e-2)- | **7.0740e-1(2.69e-2)+** | 7.6569e-1(1.28e-2)- | 6.9550e-1(2.40e-2)= | **6.9454e-1(2.51e-2)** |
| WFG6 | 10 | 6.6176e-1(4.37e-2)= | **6.1578e-1(3.80e-2)+** | 6.6015e-1(2.67e-2) | 6.9613e-1(4.35e-2)= | **6.6849e-1(5.76e-2)=** | 6.7155e-1(3.04e-2) |
| | 30 | 7.6832e-1(4.67e-2)= | **7.5581e-1(3.53e-2)+** | 7.7674e-1(2.11e-2) | 7.9686e-1(4.82e-2)- | **7.7036e-1(3.54e-2)+** | 7.8081e-1(2.73e-2) |
| | 50 | 8.2959e-1(2.50e-2)- | 8.1146e-1(2.50e-2)- | **8.0017e-1(2.39e-2)** | 8.4198e-1(4.05e-2)- | 8.2419e-1(2.99e-2)- | **8.0372e-1(2.71e-2)** |
| | 100 | 8.9024e-1(1.72e-2)- | 8.7077e-1(2.26e-2)- | **8.2709e-1(2.20e-2)** | 8.9234e-1(2.54e-2)- | 8.7694e-1(2.36e-2)- | **8.2400e-1(2.25e-2)** |
| WFG7 | 10 | 5.6903e-1(3.84e-2)= | **4.8501e-1(3.42e-2)+** | 5.2163e-1(2.11e-2) | 5.2065e-1(4.58e-2)= | **5.1756e-1(4.61e-2)+** | 5.3285e-1(2.48e-2) |
| | 30 | 6.3794e-1(2.66e-2)- | 6.1985e-1(3.01e-2)- | **5.9446e-1(1.61e-2)** | 6.2748e-1(3.09e-2)- | 6.2728e-1(3.68e-2)- | **5.9190e-1(1.79e-2)** |
| | 50 | 6.7276e-1(2.49e-2)- | 6.5438e-1(2.30e-2)- | **6.0914e-1(1.35e-2)** | 6.6321e-1(3.00e-2)- | 6.6139e-1(2.50e-2)- | **6.0562e-1(1.39e-2)** |
| | 100 | 7.0070e-1(1.85e-2)- | 6.8507e-1(1.94e-2)- | **6.2251e-1(1.45e-2)** | 6.9108e-1(1.70e-2)- | 6.8881e-1(1.28e-2)- | **6.2302e-1(1.75e-2)** |
| WFG8 | 10 | 6.8348e-1(4.23e-2)+ | **6.3647e-1(3.86e-2)+** | 7.4040e-1(3.11e-2) | 7.2561e-1(4.03e-2)= | **6.5535e-1(3.40e-2)+** | 6.6283e-1(5.02e-2) |
| | 30 | 7.1508e-1(3.90e-2)= | **6.6658e-1(2.96e-2)+** | 7.2929e-1(2.71e-2) | 7.2687e-1(2.31e-2)= | **6.7138e-1(3.35e-2)+** | 6.9754e-1(3.73e-2) |
| | 50 | 7.2910e-1(3.42e-2)- | **7.0166e-1(2.33e-2)+** | 7.1390e-1(1.77e-2) | 7.2182e-1(2.30e-2)- | 7.0529e-1(2.71e-2)= | **7.0808e-1(3.26e-2)** |
| | 100 | 7.6027e-1(2.47e-2)- | 7.2506e-1(2.52e-2)= | **7.1474e-1(2.06e-2)** | 7.3887e-1(2.34e-2)- | 7.2698e-1(2.82e-2)- | **7.0999e-1(1.96e-2)** |
| WFG9 | 10 | 5.4364e-1(7.69e-2)+ | **5.1060e-1(8.00e-2)+** | 5.9307e-1(3.99e-2) | 5.8136e-1(6.02e-2)- | **5.2266e-1(9.75e-2)=** | 5.4804e-1(8.86e-2) |
| | 30 | 7.8049e-1(7.18e-2)- | 7.5680e-1(5.44e-2)- | **7.2985e-1(3.59e-2)** | 7.7182e-1(8.81e-2)- | 8.0036e-1(8.32e-2)- | **7.3293e-1(6.16e-2)** |
| | 50 | 8.5295e-1(6.49e-2)- | 8.5317e-1(5.60e-2)- | **7.6220e-1(4.32e-2)** | 8.4950e-1(7.12e-2)- | 8.5860e-1(6.68e-2)- | **7.6224e-1(4.62e-2)** |
| | 100 | 9.2945e-1(4.19e-2)- | 9.1105e-1(3.75e-2)- | **7.7577e-1(5.19e-2)** | 9.1149e-1(6.94e-2)= | 9.2901e-1(6.03e-2)- | **7.6596e-1(6.46e-2)** |
| +/-/= | | vs. FCSEA: 10/43/11 | vs. FCSEA: 24/31/9 | ——— | vs. FREMO: 9/47/8 | vs. FREMO: 17/30/17 | ——— |

too large, the benefits of dimensionality reduction diminish, leading to increased training cost and reduced efficiency.

These findings suggest that moderate values of $k$ strike a balance between information preservation and compression, enabling efficient surrogate training without compromising accuracy. Furthermore, we are exploring adaptive strategies in which $k$ is tuned online, guided by reconstruction loss or validation performance. Such adaptive schemes could further improve scalability, especially for problems where complexity and dimensionality vary significantly. A more comprehensive investigation of adaptive latent dimensions will be left for future work.

**Summary.** Across diverse benchmarks, including high-dimensional, many-objective, and large-scale EMOPs, the proposed FSMOEA framework consistently demonstrates superior optimization performance and scalability. The empirical results reinforce our theoretical claims: embedding population context into representations and reducing search dimensionality are effective strategies for scaling surrogate-assisted evolutionary algorithms to challenging real-world optimization problems.

Table 6: Average IGD results of FCSEA and its two variants in solving many-objective MaF1-13 problems with $m \in \{5, 10\}$, $N$=50, $FE_{max}$=500.

| Problem | $m$ | $n$ | CSEA | CSEA-V1 | FCSEA |
|---|---|---|---|---|---|
| MaF1 | 5 | 14 | 2.8305e-1 (3.96e-2) + | **2.4803e-1 (2.79e-2) +** | 3.6003e-1 (4.39e-2) |
| | 10 | 19 | 4.5619e-1 (6.55e-2) - | 2.9897e-1 (6.35e-2) + | 3.8236e-1 (4.99e-2) |
| MaF2 | 5 | 14 | 9.6626e-2 (2.05e-3) - | 9.2520e-2 (1.70e-3) - | **8.0298e-2 (1.45e-3)** |
| | 10 | 19 | 3.4265e-1 (1.88e-2) - | 3.2562e-1 (1.83e-2) = | **3.1149e-1 (2.37e-2)** |
| MaF3 | 5 | 14 | 5.1804e+5 (2.24e+5)- | 4.0637e+5 (2.65e+5) - | **2.3043e+5 (3.06e+5)** |
| | 10 | 19 | 6.3067e+5 (2.36e+5)- | 6.8348e+5 (4.50e+5) - | **5.6225e+5 (3.00e+5)** |
| MaF4 | 5 | 14 | 2.7916e+3 (5.28e+2) + | **2.2574e+3 (6.13e+2) +** | 4.8622e+3 (1.18e+3) |
| | 10 | 19 | 7.4222e+4 (1.55e+4) + | **5.8497e+4 (1.57e+4) +** | 1.5364e+5 (3.99e+4) |
| MaF5 | 5 | 14 | 4.6208e+0 (8.78e-1) + | **4.5983e+0 (6.58e-1) +** | 7.5132e+0 (6.63e-1) |
| | 10 | 19 | 1.3373e+2 (1.65e+1) + | 1.1903e+2 (1.93e+1) + | 1.6103e+2 (1.29e+1) |
| MaF6 | 5 | 14 | 9.8201e+0 (3.02e+0) - | **1.9858e+0 (1.19e+0) +** | 4.4252e+0 (1.85e+0) |
| | 10 | 19 | 4.8172e+0 (2.81e+0) - | 1.0789e+0 (2.20e+0) - | **5.4427e-1 (2.50e-1)** |
| MaF7 | 5 | 24 | 1.3063e+1 (1.33e+0)- | 8.5147e+0 (1.43e+0) - | **7.0564e+0 (1.28e+0)** |
| | 10 | 29 | 2.8729e+1 (2.21e+0)- | 2.4834e+1 (2.99e+0) - | **2.1167e+1 (4.27e+0)** |
| MaF8 | 5 | 2 | 4.3217e+2 (3.55e+2) - | 5.2116e+2 (3.03e+2) - | **4.2658e+1 (4.71e+1)** |
| | 10 | 2 | 5.5070e+2 (3.06e+2) - | 5.5978e+2 (3.59e+2) - | **1.1319e+2 (1.06e+2)** |
| MaF9 | 5 | 2 | 3.5075e+2 (2.05e+2) - | 1.6467e+2 (1.37e+2) - | **2.7006e+1 (2.33e+1)** |
| | 10 | 2 | 5.3668e+2 (2.91e+2) - | 3.3435e+2 (3.10e+2) - | **2.6680e+1 (2.77e+1)** |
| MaF10 | 5 | 14 | 2.5780e+0 (8.39e-2)- | 2.1686e+0 (9.94e-2) - | **2.1004e+0 (8.59e-2)** |
| | 10 | 19 | 3.4019e+0 (4.91e-2)- | 3.1342e+0 (1.12e-1) - | **3.1102e+0 (1.62e-1)** |
| MaF11 | 5 | 14 | 9.6036e-1 (2.58e-1) + | **8.4317e-1 (1.15e-1) +** | 1.3751e+0 (3.15e-1) |
| | 10 | 19 | 3.1090e+0 (9.38e-1) + | **2.9748e+0 (9.68e-1) +** | 3.8062e+0 (7.38e-1) |
| MaF12 | 5 | 14 | 1.6884e+0 (2.16e-1) = | 1.7186e+0 (1.90e-1) - | **1.6588e+0 (5.17e-2)** |
| | 10 | 19 | 7.8457e+0 (7.05e-1) - | 7.3896e+0 (4.41e-1) - | **6.7167e+0 (2.55e-1)** |
| MaF13 | 5 | 5 | 5.7717e-1 (1.77e-1) = | **4.3700e-1 (7.47e-2) =** | 4.8199e-1 (9.08e-2) |
| | 10 | 5 | 7.5285e-1 (2.93e-1) - | **5.8685e-1 (1.10e-1) =** | 6.2505e-1 (1.42e-1) |
| +/-/= | | | 7/17/2 | 9/14/3 | ——— |

Table 7: Average IGD results of FCSEA and its four competitors in solving large-scale LSMOP1, LSMOP5, LSMOP8, and LSMOP9 problems with $m$=2, $n \in \{100, 500, 1000\}$, $N$=50, $FE_{max}$=500.

| Problems | n | KRVEA | SMSEGO | EDNARMOEA | ABSAEA | MCEAD | CSEA | FCSEA |
|---|---|---|---|---|---|---|---|---|
| LSMOP1 | 100 | 7.4599e+0(3.30e-1)- | 7.8381e+0(6.87e-1)- | 7.9255e+0(6.92e-1)- | 7.7822e+0(4.43e-1)- | 2.3723e+0(4.94e-1)- | 4.6495e+0(4.02e-1)- | **1.6938e+0(4.86e-1)** |
| | 500 | 1.0066e+1(7.68e-2)- | 9.7443e+0(2.02e-1)- | 9.8625e+0(1.91e-1)- | 9.6522e+0(2.87e-1)- | 2.7921e+0(3.12e-1)= | 8.2129e+0(6.10e-1)- | **2.4476e+0(2.50e-1)** |
| | 1000 | 1.0341e+1(1.20e-1)- | 1.0354e+1(1.41e-1)- | 1.0389e+1(2.09e-1)- | 1.0338e+1(2.36e-1)- | 3.0904e+0(6.37e-1)- | 9.4511e+0(1.03e+0)- | **2.2623e+0(1.48e-1)** |
| LSMOP5 | 100 | 1.8637e+1(9.03e-1)- | 1.8551e+1(1.05e+0)- | 1.9189e+1(1.12e+0)- | 1.8499e+1(3.38e-1)- | 6.1851e+0(1.71e+0)- | 9.6061e+0(3.00e+0)- | **3.7395e+0(6.26e-1)** |
| | 500 | 2.1515e+1(4.97e-1)- | 2.1319e+1(4.97e-1)- | 2.1601e+1(1.41e-1)- | 2.1395e+1(2.21e-1)- | 5.2394e+0(5.06e-1)= | 1.2107e+1(1.37e+0)- | **4.9862e+0(8.04e-1)** |
| | 1000 | 2.2209e+1(2.40e-1)- | 2.2288e+1(2.36e-1)- | 2.2346e+1(2.66e-1)- | 2.2202e+1(4.24e-1)- | **5.7968e+0(9.64e-1)+** | 1.4866e+1(2.70e+0)- | 7.1774e+0(9.91e-1) |
| LSMOP8 | 100 | 1.4939e+1(5.48e-1)- | 1.4617e+1(6.00e-1)- | 1.5226e+1(5.39e-1)- | 1.5652e+1(3.12e-1)- | 3.1308e+0(4.66e-1)- | 9.6533e+0(1.14e-1)- | **2.0652e+0(4.40e-1)** |
| | 500 | 1.8102e+1(2.26e-1)- | 1.8177e+1(2.19e-1)- | 1.8039e+1 (3.21e-1)- | 1.8234e+1(4.60e-1)- | 4.7245e+0(1.04e+0)- | 1.3432e+1(1.12e+0)- | **3.4462e+0(2.99e-1)** |
| | 1000 | 1.8963e+1(2.57e-1)- | 1.8931e+1(2.39e-1)- | 1.8935e+1 (1.46e-1)- | 1.8791e+1(2.31e-1)- | 4.4632e+0(6.30e-1)- | 1.7702e+1(1.67e+0)- | **3.6393e+0(6.59e-1)** |
| LSMOP9 | 100 | 3.3385e+1(1.38e+0)- | 3.3713e+1(3.40e+0)- | 3.5367e+1 (2.89e+0)- | 3.5011e+1(2.71e+0)- | 7.9247e+0(1.41e+0)- | 3.1299e+1(3.32e+0)- | **4.2533e+0(1.68e+0)** |
| | 500 | 5.0294e+1(1.41e+0)- | 4.8453e+1(2.37e+0)- | 4.9060e+1 (1.73e+0)- | 5.0184e+1(1.48e+0)- | 1.4045e+1(4.21e+0)- | 3.9439e+1(5.96e+0)- | **6.9666e+0(1.73e+0)** |
| | 1000 | 5.3422e+1 (1.19e+0)- | 5.3178e+1(1.47e+0)- | 5.3093e+1 (1.23e+0)- | 5.3208e+1(7.69e-1) - | 1.1423e+1(2.57e+0)- | 3.9415e+1(8.39e-0)- | **8.1884e+0(6.75e-1)** |
| +/-/= | | 0/12/0 | 0/12/0 | 0/12/0 | 0/12/0 | 1/9/2 | 0/12/0 | ——— |

Table 8: Sensitivity analysis of the latent dimension $k$ across representative benchmark problems (DTLZ1, DTLZ4, DTLZ7, WFG2, WFG4, WFG6, WFG8, LSMOP5, and LSMOP9). The table reports the average IGD values obtained by FSMOEA under different settings of $k$. Results show that very small $k$ (e.g., $k = 2, 3$) leads to poor reconstruction and degraded surrogate performance, while excessively large $k$ reduces the benefits of compression and increases training cost. Performance remains stable when $k \in [8, 15]$, confirming that FSMOEA is robust to a wide range of latent dimensions.

| Problem | Dimension | FCSEA-$k$ | | | | | | | | |
|---|---|---|---|---|---|---|---|---|---|---|
| | | $k = 2$ | $k = 3$ | $k = 5$ | $k = 8$ | $k = 10$ | $k = 15$ | $k = 20$ | $k = 30$ | $k = 50$ |
| DTLZ1 | 50 | 1059.4 | 1159.0 | 1099.0 | 1037.2 | 975.3 | 971.0 | 1113.1 | 1116.9 | **924.4** |
| DTLZ1 | 100 | 1552.4 | 1421.8 | 1845.3 | 1643.7 | 1680.4 | 1722.8 | 1446.2 | **1004.0** | 1849.4 |
| DTLZ4 | 50 | 1.008 | 0.982 | 1.008 | 1.028 | **0.927** | 1.045 | 1.098 | 0.999 | 1.016 |
| DTLZ4 | 100 | 0.975 | 1.093 | 1.096 | 1.010 | 0.995 | **0.975** | 1.005 | 0.983 | 1.028 |
| DTLZ7 | 50 | 9.569 | 9.595 | 9.143 | 5.537 | **4.467** | 5.968 | 9.570 | 9.498 | 9.735 |
| DTLZ7 | 100 | 9.967 | 10.171 | 9.890 | 6.201 | **6.125** | 6.278 | 8.130 | 9.670 | 10.347 |
| WFG2 | 50 | 0.796 | 0.810 | 0.819 | 0.706 | **0.665** | 0.750 | 0.743 | 0.731 | 0.761 |
| WFG2 | 100 | 0.816 | 0.786 | **0.591** | 0.818 | 0.667 | 0.696 | 0.764 | 0.752 | 0.795 |
| WFG4 | 50 | 0.613 | 0.585 | 0.595 | 0.582 | **0.522** | 0.543 | 0.624 | 0.590 | 0.645 |
| WFG4 | 100 | 0.611 | 0.609 | 0.599 | 0.582 | **0.533** | 0.607 | 0.643 | 0.593 | 0.623 |
| WFG6 | 50 | 0.833 | 0.824 | 0.843 | 0.809 | **0.800** | 0.830 | 0.853 | 0.834 | 0.838 |
| WFG6 | 100 | 0.866 | 0.858 | 0.863 | 0.823 | 0.827 | **0.824** | 0.846 | 0.859 | 0.854 |
| WFG8 | 50 | 0.761 | 0.758 | 0.756 | **0.706** | 0.714 | 0.714 | 0.749 | 0.774 | 0.762 |
| WFG8 | 100 | 0.752 | 0.757 | 0.740 | 0.727 | **0.715** | 0.716 | 0.750 | 0.755 | 0.747 |
| LSMOP5 | 500 | 9.361 | 9.697 | 6.358 | 5.815 | **4.986** | 6.443 | 7.627 | 9.344 | 9.350 |
| LSMOP5 | 1000 | 9.782 | 10.854 | 10.754 | 8.624 | **7.177** | 9.389 | 9.458 | 10.868 | 10.426 |
| LSMOP9 | 500 | 65.694 | 29.714 | 13.311 | 8.965 | **6.967** | 7.535 | 13.216 | 33.782 | 58.100 |
| LSMOP9 | 1000 | 62.126 | 31.885 | 17.252 | 9.064 | 8.188 | **7.835** | 11.631 | 41.740 | 59.657 |

Table 9: The actual average running (seconds: s) time of each algorithm in solving the DTLZ and WFG problems: except for the KRVEA which uses the Kriging model, whose running time is significantly better, the FCSEA is comparable to other methods and is faster than CSEA.

| Problem | M | D | DirHVEI | KRVEA | LDSAF | MCEAD | SFADE | CSEA | FCSEA |
|---|---|---|---|---|---|---|---|---|---|
| DTLZ2 | 3 | 100 | 8.0262e+1(3.37e+1) | **1.7069e-2(4.62e-3)** | 3.9912e+1(4.57e+0) | 1.2708e+1(5.77e-1) | 6.3759e+1(7.86e+0) | 7.3960e+1(8.44e+0) | 1.3261e+1(1.50e+0) |
| DTLZ4 | 3 | 100 | 8.0929e+1(3.41e+1) | **1.3602e-2(2.83e-3)** | 3.6510e+1(4.79e+0) | 1.0395e+1(6.71e-1) | 6.3768e+1(8.07e+0) | 6.1916e+1(1.04e+1) | 1.3306e+1(1.93e+0) |
| DTLZ7 | 3 | 100 | 8.3704e+1(3.42e+1) | **2.5250e-2(7.90e-3)** | 3.5902e+1(4.28e+0) | 1.1958e+1(1.13e+0) | 6.3741e+1(5.82e+0) | 5.8979e+1(5.45e+0) | 1.5382e+1(1.84e+0) |
| WFG1 | 3 | 100 | 8.1794e+1(3.41e+1) | **3.1586e-2(9.78e-3)** | 3.6044e+1(3.79e+0) | 1.0209e+1(3.66e-1) | 5.9050e+1(5.39e+0) | 5.3880e+1(4.86e+0) | 9.0869e+0(1.02e+0) |
| WFG5 | 3 | 100 | 8.0870e+1(3.37e+1) | **1.6300e-2(3.20e-3)** | 3.4993e+1(3.69e+0) | 1.0160e+1(3.71e-1) | 6.3830e+1(8.30e+0) | 5.265 1e+1(4.27e+0) | 8.0373e+0(7.47e-1) |
| WFG8 | 3 | 100 | 8.0275e+1(3.32e+1) | **2.8541e-2(7.96e-3)** | 3.4811e+1(3.74e+0) | 1.0403e+1(2.83e-1) | 6.7994e+1(9.47e+0) | 5.3162e+1(4.96e+0) | 7.4833e+0(8.99e-1) |

All source codes were implemented on the PlatEMO, and all experiments were conducted on a personal computer equipped with an Intel Core i5-10505 CPU (3.2 GHz) and 24 GB of RAM.

