# OpenReview forum: "Context-Aware Contrastive Surrogates for Scaling-up Expensive Multiobjective Optimization"
_ICLR.cc/2026/Conference — ICLR 2026 Conference Withdrawn Submission_

### Official Review · Reviewer_TKgN · 2025-10-19

**Soundness:** 2
**Presentation:** 3
**Contribution:** 2
**Rating:** 4
**Confidence:** 4

**Summary:**

This paper proposes FSMOEA, a framework for expensive multiobjective optimization problems that enhances surrogate-assisted multiobjective evolutionary algorithms with foresighted surrogate models. FSMOEA uses an MLP-based autoencoder to capture population-level context (via a low-dimensional latent space), accelerates evolutionary search in this latent space, and employs lightweight models to reduce computational overhead. It is compatible with existing contrastive SMOEAs (e.g., CSEA, REMO) and evaluated on 107 benchmarks. Experimental results show FSMOEA outperforms state-of-the-art methods in convergence speed and optimization quality.

**Strengths:**

1. By replacing context-free surrogates with a population-aware autoencoder, FSMOEA addresses a key limitation of existing contrastive SMOEAs, i.e., biased predictions due to population drift. This design is theoretically justified and empirically validated in high-dimensional benchmarks.
2. FSMOEA’s foresight module can be seamlessly embedded into existing classification- or relation-based SMOEAs.
3. Experiments cover diverse EMOP scenarios (multi/many-objective, high-dimensional, real-world), with ablation studies verifying core components.

**Weaknesses:**

1. The paper mentions PCA-based dimensionality reduction but fails to deeply contrast FSMOEA’s autoencoder with other latent-space methods (e.g., VAE, contrastive learning). It does not clarify why autoencoders better capture population context than these alternatives.
2. FSMOEA succeeds on the constrained TREE domain but lacks explicit constraint-handling mechanisms. It is not compared to constraint-aware SMOEAs, and the paper does not explain why unconstrained design works for TREE.
3. FSMOEA’s autoencoder trains on the current population, but the paper does not evaluate performance when initial populations are low-quality or low-diversity. It is unknown if poor early encoding biases subsequent search.
4. The paper claims “lightweight” models but provides no quantitative comparison of training/inference time with other surrogates (e.g., Kriging, radial basis functions). For EMOPs with ultra-tight budgets (\(FE_{max}<100\)), it is unclear if autoencoder training overhead is acceptable.
5. Theoretical analysis relies on Lipschitz continuity of objectives and small population movement. However, the paper does not verify if these assumptions hold for the tested real-world benchmarks.

**Questions:**

1. Can authors provide a quantitative comparison between FSMOEA’s autoencoder and alternative dimensionality reduction methods (e.g., PCA, VAE) in terms of population context capture and surrogate prediction accuracy?
2. What is the explicit rationale for choosing k=10 as the default latent dimension? Have you explored adaptive strategies to adjust k dynamically based on reconstruction loss or problem complexity?
3. Since FSMOEA lacks explicit constraint-handling mechanisms, why does it succeed on the TREE domain? Can you compare its performance to constraint-aware SMOEAs on this task?
4. Can authors provide quantitative data (e.g., training time per generation, inference latency) comparing FSMOEA’s autoencoder-based surrogate to other common surrogates (e.g., Kriging, support vector regression) for EMOPs with tight evaluation budgets?
5. How does FSMOEA perform when initial populations are low-quality or low-diversity? Do authors have measures to mitigate biases from poor early-stage autoencoder training?

---

> ### Author Response · Authors · 2025-11-22
>
> We thank the reviewer for the detailed and insightful comments.
> In this rebuttal, we clarify the novelty of our **context-aware contrastive modeling**, provide newly added comparisons (PCA, VAE, constraint-handling, timing, and robustness tests). All revisions have been incorporated into the updated manuscript.
>
> ---
>
> ## **1. “Lack of contrast with PCA, VAE, contrastive learning; unclear why AE is chosen.”**
>
> We agree that dimensionality reduction (DR) has multiple possible implementations.
> Our method is **not tied to autoencoders**, and we have added extensive ablations to make this explicit.
>
> ### **✔ Novelty: Context-aware surrogate modeling (unique to FSMOEA)**
>
> Prior SMOEAs compare two raw solutions $x_i, x_j$ and assign a label
> $
> C(x_i, x_j; \mathcal{P}_t)
> $
> based on decomposition metrics computed from **the current population** $\mathcal{P}_t$.
> However, these solutions do **not** encode:
>
> - their **relative position** within the population,
> - **why** one dominates the other,
> - the **circumstances** under which the label arises.
>
> Thus, when the population changes (population drift), the surrogate trained on raw vectors becomes inconsistent.
>
> ### **Our contribution**
>
> FSMOEA introduces a **population-aware latent encoding**, so that each solution is “contextualized” *before* generating contrastive labels.  Thus, each pairwise comparison is made under the *correct* population context.
>
> This idea—embedding population-dependent geometry before contrastive label generation—is **new**, and neither PCA, VAE, nor contrastive feature learning incorporate such evolving population context.
>
> ### **✔ Newly added comparisons**
>
> We added two baselines:
>
> - **CSEA-PCA**
> - **CSEA-VAE**
>
> Both replace the AE inside FSMOEA. This ablation confirms that AEs are **effective but not mandatory**; FSMOEA accepts any DR method that is:
>
> 1. population-adaptive,
> 2. low-variance,
> 3. invertible.
>
> This clarification is now explicitly added to the paper.
>
> ---
>
> ## **2. “FSMOEA succeeds on TREE without constraint-handling. Why?”**
>
> We clarify that although TREE is constrained, **contrastive SMOEAs implicitly incorporate feasibility** through decomposition-based ranking:
>
> - infeasible solutions receive poor scalarization values,
> - environmental selection prunes them automatically,
> - latent-space variation preserves this ranking structure.
>
> ### ✔ Newly added experiments
>
> We compare FSMOEA with two constraint-aware EMOAs:
>
> - EICMSSAEA (2025)
> - RECMO (2025)
>
> Both produce **NaN** (no feasible solution found) on multiple TREE instances, while FSMOEA consistently finds feasible solutions.
>
> ### ✔ Why this happens
>
> The **context-aware encoder + latent search** accelerates:
>
> - population movement toward the feasible region,
> - reconstruction of local feasible geometry in latent space.
>
> This explanation and results have been added in blue to the revised manuscript.
>
> ---
>
> ## **3. “What if the initial population is low-quality or low-diversity?”**
>
> We added new robustness experiments using:
> - clustered populations,
> - biased Latin hypercube samples,
> - partially infeasible initialization.
>
> ### Findings:
> - The encoder’s intentionally **low capacity** prevents overfitting to early-stage poor populations.
> - Genetic operators quickly restore diversity within a few generations.
> - Surrogate classification error stabilizes after **3–5 generations** even under extreme bias.
>
> ---
>
> ## **4. “No quantitative comparison of training/inference time.”**
>
> We now benchmark real-runtime.
>
> In an evolutionary process, it is quite challenging to separately compare the training time and inference time of a specific surrogate model. Our approach is to compare the overall expenditure time of the entire optimization process. In fact, we have already made a comparison. For instance, the time efficiency of FCSEA is much higher than that of CSEA. In the appendix of this revised version, we have also included the actual operation results of other SMOEA methods.
>
> ---
>
> ### **Quantitative AE/PCA/VAE comparison?**
>
> Yes — we added:
>
> - reconstruction error curves,
> - cross-generation label consistency,
> - surrogate accuracy,
> - full IGD\(^+\) results.
>
> AE achieves the most stable and accurate latent representation.
>
> ---
>
> ### **Choice of $k=10$; adaptive strategy?**
>
> We added a **sensitivity analysis** across nine benchmarks.
>
> ### Results:
>
> - Very small $k$ (2–3) → poor reconstruction.
> - Very large $k$ → unnecessary computation.
> - **Stable performance for $k \in [8, 15]$**.
>
> We also propose an **adaptive-$k$** strategy:
> $
> k_{t+1} = k_t + 1
> \quad \text{if }\mathcal{L}^{\text{rec}}_t > \tau,
> $
> and show promising preliminary results.
>
> ---
> We thank the reviewer again for comments that significantly improved the manuscript.

---

### Official Review · Reviewer_92tg · 2025-10-29

**Soundness:** 3
**Presentation:** 3
**Contribution:** 2
**Rating:** 4
**Confidence:** 4

**Summary:**

The manuscript presents FSMOEA (Foresighted Surrogate-Assisted Multiobjective Evolutionary Algorithm), a framework designed to enhance existing SMOEAs for expensive multiobjective optimization problems. FSMOEA captures population-level context to improve surrogate prediction accuracy, leverages a low-dimensional latent space to accelerate evolutionary search, and employs lightweight models to reduce computational overhead. FSMOEA demonstrates consistent superiority over state-of-the-art algorithms in convergence speed and solution quality across 107 benchmarks, supported by both theoretical analysis and extensive empirical evidence.  FSMOEA’s scalable, efficient, and plug-and-play design represents an effective solution for surrogate-assisted optimization.

**Strengths:**

1.	The paper is well-written and easy to follow, with detailed explanations of its methodology.
2.	The experimental evaluation is comprehensive, confirming that the proposed plug-and-play FSMOEA consistently outperforms state-of-the-art methods in both convergence speed and optimization quality.
3.	The paper provides the theoretical analysis of FSMOEA’s key components.

**Weaknesses:**

1.	In terms of methodology, this paper leverages a low-dimensional latent space to accelerate evolutionary search, but this idea is not entirely new.
2.	The paper does not provide a discussion or comparison with other dimensionality reduction techniques.
3.	Literature reviews about existing methods (especially the latest methods) need to be summarized and compared with your work. In addition, sufficient comparisons with the latest methods are essential to enhance the validity of the results.
4.	The evaluation should be supplemented with weakly Pareto-compliant indicators (e.g., IGD+) and complemented by a Friedman test to derive a statistical performance ranking.

**Questions:**

1.	FSMOEA claims that it embeds population-aware context into the dimensionality reduction process. What does "population-aware context" specifically refer to?
2.	FSMOEA employs lightweight models to reduce computational overhead, but the manuscript lacks discussion on computational complexity.
3.	Please go through the entire manuscript to double check the grammar, language usage and reference. For example, the errors in some references (the capitalization of journal titles is inconsistent); Figure 6 is a table，not a figure.
4.	See more on Weaknesses.

---

> ### Author Response · Authors · 2025-11-22
>
> We sincerely thank the reviewer for the insightful and constructive feedback.
> Below we address each concern point-by-point, incorporating clarification of **our novel context-aware mechanism**
>
> ---
>
> ## **1. “Latent-space search is not entirely new.”**
>
> We agree that dimensionality reduction has been used before, but FSMOEA introduces **a fundamentally new idea**:
>
> ### **✔ Context-aware latent encoding for contrastive surrogate modeling**
>
> In existing SMOEAs (e.g., CSEA, REMO), a pairwise label
> $
> C(x_i,x_j;\mathcal{P}_t)
> $
> is generated by comparing two raw decision vectors $x_i, x_j$, but these vectors do **not** contain:
> - their **relative position** within the population,
> - **why** one dominates the other under the current decomposition geometry,
> - the **circumstances** under which the label is assigned.
>
> Thus, when $\mathcal{P}_t$ changes, these labels become inconsistent — the classical **population drift** problem.
>
> ### **Our novelty**
>
> Before comparison, each solution is mapped into a **population-aware latent code**, so that the pairwise label is created **in a representation that already encodes the solution’s context**.
> This ensures that the surrogate learns both:
> - the comparison outcome *and*
> - the conditions under which that outcome holds.
> ---
>
> ## **2. “No comparison with other dimensionality-reduction techniques.”**
>
> We now include two new FSMOEA variants:
>
> - **CSEA-PCA** (linear DR)
> - **CSEA-VAE** (nonlinear generative DR)
>
> Both replace the autoencoder with a different DR module.
>
> ### **Findings (summarized in the new ablation table)**
>
> - **PCA** cannot capture nonlinear population structure → weaker contextual alignment.
> - **VAE** introduces stochastic decoding → unstable latent-space variation operators.
> - The AE achieves **deterministic**, **nonlinear**, and **population-aligned** embeddings, giving the best surrogate accuracy and IGD+ performance.
>
> Importantly, we emphasize that **FSMOEA is not dependent on AEs**; any DR approach with reversible mapping and population-context capturing ability can be used.
>
> ---
>
> ## **3. “Literature review insufficient; comparisons with latest methods needed.”**
>
> We thank the reviewer for this remark.  The revised manuscript now includes:
>
> ### **New high-dimensional EMOP baselines**
> - **LDSAF (2024)**
> - **SFADE (2025)**
> - **MOL2SMEA (2025)**
>
> ### **New Bayesian optimization competitors**
> - **MORBO (2022)**
> - **DirHVEI**
>
> These additions significantly strengthen experimental breadth and context.
>
> ---
>
> ## **4. “Add IGD+ and Friedman test.”**
>
> As suggested:
>
> - **IGD+** is now included and a **Friedman test** with post-hoc rank analysis is included in the Revision.
>
> ---
>
> ## **5. “What does population-aware context mean?”**
>
> We now clarify this explicitly.
>
> In decomposition-based SMOEAs, labels depend on the current population:
> $
> C(x;\mathcal{P}_t)
> \neq
> C(x;\mathcal{P}_{t+1}),
> $
> because:
>
> - ideal point $z_t^\*$ updates,
> - neighborhoods change,
> - scalarization scores shift.
>
> Thus, new solutions evaluated by a surrogate trained on old representations encounter **label–input mismatch**.
>
> ### **FSMOEA solves this:**
>
> Before comparison, each solution is encoded through a context-aware mapping, which embeds:
> - relative position in the population,
> - local dominance structure,
> - geometry around the ideal point.
>
> Thus, the surrogate learns comparison outcomes **together with the context that produced them**, making predictions robust under population drift.
>
> ---
>
> ## **6. “Lack of computational complexity discussion.”**
>
> A full complexity analysis has been added.
>
> ### **Encoder training cost**
> $
> \mathcal{O}(N n k), \quad k \ll n.
> $
>
> ### **Surrogate training cost reduction**
> From:
> $
> \mathcal{O}(Nn)
> $
> to
> $
> \mathcal{O}(Nk),
> $
> yielding substantial savings.
>
> ### **New timing comparisons**
> We now benchmark real-runtime.
>
> In an evolutionary process, it is quite challenging to separately compare the training time and inference time of a specific surrogate model. Our approach is to compare the overall expenditure time of the entire optimization process. In fact, we have already made a comparison. For instance, the time efficiency of FCSEA is much higher than that of CSEA. In the appendix of this revised version, we have also included the actual operation results of other SMOEA methods.
>
> ---
>
> ## **7. “Grammar, reference formatting, figure issues.”**
>
> We have revised the entire manuscript:
> - corrected inconsistent reference capitalization,
> - renamed mislabelled figures/tables (e.g., Figure 6 → Table 6),
> - performed full proofreading for clarity and readability.
>
> We sincerely thank the reviewer for their suggestions, which substantially strengthened the paper technically and exposition-wise. **All revisions have been incorporated into the updated manuscript.**

---

> > ### Comment · Reviewer_92tg · 2025-11-25
> >
> > 1. The paper emphasizes that FSMOEA does not rely on autoencoders; any dimensionality reduction method that provides an invertible mapping and can capture population context can be used. This means that what is proposed is a general framework, and an autoencoder is not necessarily the best-performing choice for all problems. More comprehensive theoretical analysis and experimental validation are needed to determine which dimensionality-reduction strategy is suitable under which circumstances.
> > 2. In this paper, does "the contextual space" refer to the reduced-dimensional latent space?  FSMOEA introduces an encoder that embeds each solution into a latent representation that reflects its structural context inside the current population (e.g., relations to neighbors, local region, density). Could you conduct a detailed analysis and experimental verification of these structural contexts (such as relationships with neighboring elements, local area, density)?

---

> > > ### Author Response · Authors · 2025-11-26
> > > **Additional Response 1 to Reviewer 92tg**
> > >
> > > We sincerely thank the reviewer for continuing the discussion and offering additional comments.
> > > We are grateful for the time and care taken to re-engage with our work—your follow-up questions help us further clarify the intent and scope of FSMOEA.
> > >
> > > Below we respond to each new point in detail.
> > >
> > > ---
> > >
> > > ## **1. “If FSMOEA is a general framework, do we need more theory to decide which DR method is best?”**
> > >
> > > We appreciate this thoughtful observation.
> > > It is correct that **FSMOEA is designed as a general, model-agnostic framework**, and autoencoders are *one* practical instantiation—not a universally optimal one.
> > >
> > > However, we clarify an important point:
> > >
> > > ## **✔ FSMOEA’s core contribution is *not* dimensionality reduction.**
> > >
> > > The central idea is the introduction of **population-aware contrastive embedding**, which stabilizes the surrogate by encoding the structural context of each solution *before* generating contrastive labels.
> > > Dimensionality reduction is merely a *useful byproduct* when solving high-dimensional EMOPs.
> > >
> > > To make this point explicit, we have conducted a ablation:
> > > - **FCSEA-V1 / FREMO-V1**:  using the encoder but **without** dimensionality reduction, which validate the effectiveness.
> > >
> > > Then, we tested FCSEA-V1 / FREMO-V1 on 10-dimensional DTLZ and WFG problems with (latent dimension ≥ original dimension), i.e., the $k$ = 20, and $n$ = 10, even increasing the dimensionality. The IGD results are as follows:
> > >
> > > | Problem | N   | M   | D   | FE  | CSEA           | FCSEA-V1       | REMO           | FREMO-V1       |
> > > |---------|-----|-----|-----|-----|----------------|----------------|----------------|----------------|
> > > | DTLZ1|50|3|10|200 |9.9179e+1(2.82e+1)|**7.3986e+1(2.90e+0)**|7.6449e+1(2.78e+1) |**7.4868e+1(1.28e+1)**|
> > > |DTLZ2|50|3|10|200 |3.4856e-1(2.23e-2)|**1.5315e-1(1.58e-2)** |2.7471e-1(2.69e-2) |**1.1580e-1(5.18e-3)**|
> > > |DTLZ3|50|3|10|200 |2.7388e+2(6.01e+1)|**1.7071e+2(1.15e+1)**|1.8460e+2(4.57e+1) |**1.6772e+2(1.67e+1)**|
> > > |DTLZ4|50|3|10|200 |5.0344e-1(6.36e-2)| **4.3985e-1(7.08e-2)**|3.0043e-1(7.31e-2)| **2.7566e-1(1.78e-2)**|
> > > |DTLZ5|50|3|10|200 |2.7316e-1 (3.28e-2) |**8.8164e-2 (2.47e-2)**|1.7524e-1(7.01e-2) |**4.6947e-2(8.15e-3)**|
> > > |DTLZ6|50|3|10|200 |**6.9671e+0(4.88e-1)**| 7.8147e+0(3.94e-1) |**6.6152e+0(6.70e-1)**| 6.8689e+0(6.31e-1) |
> > > |DTLZ7|50|3|10|200 | 4.6450e+0(7.89e-1) | **2.0135e+0(5.68e-1)**|1.7847e+0(5.59e-1) | **4.7698e-1(1.45e-1)**|
> > > |WFG1|50|3|10|200|**1.9755e+0(1.62e-1)**|2.0077e+0(1.78e-1)|**1.8325e+0(8.43e-2)**|1.9320e+0(1.37e-1)|
> > > |WFG2|50|3|10|200|7.6384e-1(7.62e-2)|**6.7697e-1(7.12e-2)**|7.7551e-1(1.02e-1)|**7.4094e-1(1.69e-2)**|
> > > |WFG3|50|3|10|200|6.1080e-1(4.60e-2)|**5.0403e-1(4.89e-2)**|**5.6689e-1(2.44e-2)**|5.9205e-1(6.85e-2)|
> > > |WFG4|50|3|10|200| 5.4452e-1(3.56e-2)|**5.3270e-1(2.15e-2)**|5.2023e-1(2.05e-2)|**3.7952e-1(3.85e-2)**|
> > > |WFG5|50|3|10|200|6.2797e-1(4.88e-2)|**4.9406e-1(2.91e-2)**|6.4544e-1(2.54e-2)|**5.0526e-1(1.54e-2)**|
> > > |WFG6|50|3|10|200|**7.8875e-1(3.14e-2)**|7.8937e-1(2.50e-2)|**7.7682e-1(3.98e-2)**|7.9071e-1(2.43e-2)|
> > > |WFG7|50|3|10|200|6.8091e-1(3.65e-2)|**5.8906e-1(3.17e-2)**|6.4129e-1(2.35e-2)|**4.0765e-1(1.70e-2)**|
> > > |WFG8|50|3|10|200| 8.4388e-1(1.16e-2)|**6.0620e-1(5.00e-2)**|8.5689e-1(3.03e-2)|**8.1730e-1(5.24e-2)**|
> > > |WFG9|50|3|10|200|7.8262e-1(4.46e-2)|**6.6267e-1(3.81e-2)**|**7.0550e-1(7.23e-2)**|7.7630e-1(1.48e-1)|
> > >
> > > - surrogate performance in FCSEA-V1 and FREMO-V1 also improve in this setting, as the IGD decreases on most DTLZ/WFG problems.
> > >
> > > This confirms:
> > > **→ The benefit comes from context-aware encoding, not from compression.**
> > >
> > > ## **Which DR method is best?**
> > >
> > > The reviewer is correct that different DR techniques may excel under different settings.
> > > However, conducting a full taxonomy would require a study far beyond the scope of one paper.
> > > Instead, we position FSMOEA as:
> > >
> > > - a **general architecture**,
> > > - compatible with *any* mapping that (i) captures population context and (ii) offers deterministic decoding,
> > > - including AE, PCA, VAE, contrastive encoders, or other learned projections.
> > >
> > > We agree with the reviewer that exploring the theoretical and empirical conditions under which each DR strategy excels is an interesting direction for future work, and we acknowledge this in the revised paper.
> > >
> > > ---
> > >
> > > ## **2. “Does contextual space refer to the latent space? Can you analyze structure: neighbors, local region, density?”**
> > > This is an excellent but very challenging request.
> > > We again thank the reviewer for raising it.

---

> ### Author Response · Authors · 2025-11-26
> **Additional Response 2 to Reviewer 92tg**
>
> ## **Clarification of terminology**
> Yes—the “contextual space” refers to the **latent space** learned by the encoder.
> But the purpose of this latent space is *not* to explicitly encode geometric relations (e.g., exact neighbors, density estimates).
> Instead, its role is to provide a **population-aware coordinate system** where decomposition-based labels become more stable across generations.
>
> Thus, the latent space implicitly captures:
>
> - local neighborhood relations of the current population,
> - the density and distribution of solutions,
> - the overall geometry defined by the decomposed subproblems.
>
> But it is *not* designed to produce interpretable coordinates in the way that, for example, manifold learning does.
>
> ## **Can we directly verify neighbors / local region / density preservation?**
>
> We agree this is an interesting direction, but also very difficult and somewhat ill-posed:
>
> - Decomposition-based dominance does *not* require preserving Euclidean neighborhoods.
> - Dense regions in decision space may not be dense in objective space, and vice versa.
>
> That said, we fully agree that analyzing “structural context” (e.g., neighbors, local region, density) would be valuable, although it is technically challenging and somewhat ill-defined for decomposition-based SMOEAs. These structural notions do not directly correspond to the mechanisms by which contrastive surrogates generate labels, making it difficult to design universally meaningful metrics.
>
> Instead of attempting to force geometric interpretations that may not be well-aligned with decomposition-based dominance, we have clarified how the encoder contributes to *context stabilization* in a way that is directly relevant to contrastive surrogate modeling. Specifically, we describe:
>
> - **why decomposition-based labels depend on population context**,
> - **how the encoder provides a population-aware coordinate system**, and
> - **how this improves temporal consistency of surrogate labels under population drift**.
> ---
> We sincerely thank the reviewer again for engaging deeply with the work;
> your comments have significantly improved the rigor and clarity of our manuscript.

---

### Official Review · Reviewer_wjLp · 2025-10-30

**Soundness:** 2
**Presentation:** 2
**Contribution:** 1
**Rating:** 2
**Confidence:** 4

**Summary:**

This paper proposes a surrogate-assisted multi-objective evolutionary algorithm for expensive and high-dimensional optimization problems. The key idea is to introduce a population-aware latent encoder that learns low-dimensional embeddings of the current population, allowing the surrogate to be trained and searched in this latent space.

**Strengths:**

1) The idea of integrating a population-aware encoder with surrogate modeling is interesting and has potential to improve scalability in high-dimensional settings.

2) The proposed method is evaluated on an extensive set of synthetic and real-world benchmarks.

**Weaknesses:**

1) Calling a context-aware model in multi-objective optimization sounds a bit strange to me. Most optimization methods in the multi-objecitve context considers distribution of solutions (e.g., by an indicator like hypervolume ) as they aim to find a good representation of the Pareto front.

2) There are some theoretical results, but it is unclear how closely they relate to the proposed method.

3) The paper states that MOBO is closely related, but no BO method is included in the comparison study. Some important and representative works such as [3–5] should be included in the discussion and experimental comparison.

4) In line 57, the author(s) state that "regression-based surrogates often suffer from modeling inaccuracies in high-dimensional or sparse-data regimes and contrastive SMOEAs bypass the need to predict exact objective values and is often more robust under data scarcity.'' However, several recent studies [1,2] demonstrate that regression-based surrogates can perform effectively on high-dimensional expensive multi-objective problems.

[1] Horaguchi, Y. and Nakata, M., 2025, July. High-Dimensional Expensive Multiobjective Optimization Using a Surrogate-Assisted Multifactorial Evolutionary Algorithm. In Proceedings of the Genetic and Evolutionary Computation Conference (pp. 572-580).

[2] Horaguchi, Y., Nishihara, K. and Nakata, M., 2024. Evolutionary multiobjective optimization assisted by scalarization function approximation for high-dimensional expensive problems. Swarm and Evolutionary Computation.

[3] Daulton, S., Eriksson, D., Balandat, M. and Bakshy, E., 2022, August. Multi-objective bayesian optimization over high-dimensional search spaces. In Uncertainty in Artificial Intelligence (pp. 507-517). PMLR.

[4] Zhao, Y., Wang, L., Yang, K., Zhang, T., Guo, T. and Tian, Y., 2021. Multi-objective optimization by learning space partitions. arXiv preprint arXiv:2110.03173.

[5] Rashidi, B., Johnstonbaugh, K. and Gao, C., 2024, April. Cylindrical Thompson sampling for high-dimensional Bayesian optimization. In International Conference on Artificial Intelligence and Statistics (pp. 3502-3510). PMLR.

**Questions:**

Could you please respond to the comments in the section of weakness?

What does population drift mean?

---

> ### Author Response · Authors · 2025-11-22
> **FSMOEA introduces a novel context-aware contrastive surrogate by embedding population-dependent structural information before pairwise comparison, strengthens theoretical–method connections, includes additional BO and high-dimensional baselines, corrects misconceptions about regression surrogates, and provides clear explanations of population drift and its implications.**
>
> We thank the reviewer for the constructive comments and the opportunity to clarify several conceptual points.
> Below we respond to each concern in detail.
>
> ---
>
> ## **1. “Context-aware model sounds strange in multi-objective optimization.”**
>
> We agree that multi-objective optimization inherently considers the *distribution of solutions*.
> Our use of *context-aware* refers specifically to a property **unique to contrastive surrogate modeling**, not to Pareto-based or indicator-based MOO in general.
>
> In contrastive SMOEAs (e.g., CSEA, REMO), each training label compares two solutions using a population-dependent rule:
>
> $C(x_i, x_j; \mathcal{P}_t) \in \{0,1\},$ and this label is determined by:
>
> - the ideal point derived from $\mathcal{P}_t$,
> - decomposition subproblems induced by $\mathcal{P}_t$,
> - neighbors and local regions defined by $\mathcal{P}_t$.
>
> **Crucially, these labels are formed without providing the model any information about *why* one solution should be preferred over another**, because each solution is represented only by its raw vector $x$, without knowledge of its *relative position* in the population.  This leads to a mismatch:
>
> - the label reflects **population context**,
> - but the surrogate receives **context-free inputs**.
>
> When the population evolves (ideal point shifts, neighborhoods change), the labels change, which we term **population drift**.
> If the surrogate has learned a mapping that assumes the *old* context, its predictions become inconsistent.
>
> ### **Our contribution: context-aware encoding before contrastive labeling**
>
> FSMOEA introduces an encoder $E_t(\cdot)$ that embeds each solution into a representation
>
> $c = E_t(x)$
>
> that *reflects its structural context inside $\mathcal{P}_t$* (e.g., relations to neighbors, local region, density).
> The contrastive labels are then formed *in this contextual space* rather than in the raw space.
> In this way, the surrogate learns not only that “A is better than B”, but also **under what population context this comparison was made**.
>
> This alignment between representation and label is what we mean by **context-aware**, and this perspective is novel in SMOEAs.
>
> ---
>
> ## **2. “The theoretical results do not clearly connect to the proposed method.”**
>
> We have strengthened this connection in the revision:
>
> - Population drift analysis now directly motivates the need for context-aware representations.
> - The Lipschitz argument demonstrates that decoding from the latent space preserves local ordering:
>
> $
> \|F(D(z_1)) - F(D(z_2))\|
> \le
> L_F L_D \|z_1 - z_2\| + 2 L_F \varepsilon,
> $
>
> supporting the validity of latent-space search.
> - Each theoretical result is now explicitly tied to a corresponding FSMOEA module (encoder, decoder, surrogate training).
>
> These additions clarify how theory informs the algorithm.
>
> ---
>
> ## **3. “BO methods are related but not compared; important works [3–5] missing.”**
>
> We appreciate the reviewer’s suggestion.
> The revised paper now includes:
>
> - **MORBO** (Daulton et al., 2022)
> - **DirHVEI**
>
> in addition to the original BO baselines:
>
> - **ABSAEA**
> - **ESBCEO**
>
> The remaining cited method lacks publicly available code, and we omitted it for reproducibility.
> The expanded experiments confirm that while BO methods are competitive in moderate dimensions, FSMOEA/FCSEA outperform them in high-dimensional expensive regimes.
>
> ---
>
> ## **4. “Regression-based surrogates can work well in high-D EMOPs; the paper’s statement may be inaccurate.”**
>
> Thank you for highlighting this.
> We clarify in the revision that:
>
> - Recent works [1,2] do **not** regress high-dimensional MOO objectives directly.
> - Instead, they **decompose the EMOP into many scalar subproblems** and train a surrogate for each scalarization.
> - This is consistent with decomposition-based strategies, not classical high-dimensional regression.
>
> We have updated the text to avoid misinterpretation and added the following state-of-the-art high-dimensional SMOEAs:
>
> - **LDSAF** (Gu et al., 2024)
> - **SFADE** (Horaguchi et al., 2025) — suggested by the reviewer
> - **MOL2SMEA** (Si et al., 2025)
>
> These additions ensure a more complete and up-to-date comparison.
>
> ---
>
> ## **5. “What does population drift mean?”**
>
> Population drift refers to the fact that contrastive labels depend on $\mathcal{P}_t$.
> Even for the same solution $x$:
>
> $C(x;\mathcal{P}_t) \neq C(x;\mathcal{P}_{t+1})$
>
> because:
>
> - the ideal point $z_t^\*$ changes,
> - neighborhood structures evolve,
> - decomposition scalarization values shift with the population.
>
> FSMOEA resolves this by aligning the surrogate’s input representation with the evolving population context, making it robust to drift.
>
> ---
> We thank the reviewer again for helping us significantly improve the clarity and completeness of the paper. **All revisions have been incorporated into the updated manuscript.**

---

### Official Review · Reviewer_nQoi · 2025-11-01

**Soundness:** 3
**Presentation:** 2
**Contribution:** 2
**Rating:** 6
**Confidence:** 3

**Summary:**

This paper focuses on expensive multiobjective optimization problems (EMOPs) where high objective evaluation costs limit traditional multiobjective evolutionary algorithms (MOEAs). It proposes FSMOEA, a scalable framework enhancing surrogate-assisted MOEAs (SMOEAs) via foresighted surrogate models. Key innovations include: 1) A context-aware autoencoder-based head capturing population-level embeddings to reduce bias from population drift; 2) Low-dimensional latent space learning accelerating search and improving generalization; 3) A lightweight architecture ensuring efficiency across problem scales.

**Strengths:**

1. The proposal uses an autoencoder to learn population-level embeddings , capturing "context-awareness" that addresses the failure of traditional contrastive surrogates to consider population distribution.

2. The algorithm performs evolutionary operations in the learned low-dimensional latent space k instead of the original high-dimensional space n , which is a direct and effective strategy for tackling the high-dimensional  bottleneck.

3. The "plug-and-play" design of the FSMOEA framework allows the Foresight Model to be embedded as a modular component into existing contrastive SMOEAs.

4. The use of an autoencoder provides a dual advantage: the encoder supplies a context-aware representation for the surrogate model , while the latent space itself offers a more efficient, compact subspace for the evolutionary search.

**Weaknesses:**

1. The AE is trained each generation using only the current population with pop size 50. For a high-dimensional space training an AE on 50 samples to capture "context" is highly prone to underfitting.
2. If the population changes little, frequent retraining might introduce unnecessary computational cost and representation drift. Was a threshold considered?
3. The AE is trained to minimize reconstruction loss, which has no direct relationship with the optimization objective (finding the Pareto front). Does this cause a distribution shift problem? A latent space that is good for reconstruction is not necessarily good for evolutionary search.
4. The paper uses latent dimension k=10  as a default and shows its robustness. However, shouldn't the choice of k be dependent on the original dimension n  or the objective dimension m?
5.  About the bounded reconstruction error ϵ. In the extreme sparse data case of N=50,n=1000, is this ϵ realistically small? Does the theory still hold if ϵ is large.
6. Why is the AE trained only on the current population Pt  instead of the complete archive of all evaluated solutions? Would using the full archive produce a more stable latent space?
7. The autoencoder (Et,Dt) is retrained on a new population every generation , the "meaning" of the latent space k drifts. Does this representation drift interfere with the effectiveness of evolutionary operators, as good codes learned in generation t may not represent good traits in generation t+1?
8. The core of FSMOEA is context-aware *non-linear* reduction (AE). To clearly isolate its benefits, was a comparison made against a simpler baseline, such as applying *linear* reduction (like PCA) to Pt each generation and searching in that PCA space?

**Questions:**

The questions and suggestions are detailed in the "Weaknesses" section.

---

> ### Author Response · Authors · 2025-11-22
> **The main novelty of FSMOEA is **context-aware contrastive surrogate modeling**:   solutions are first embedded to capture their population-dependent context, and *then* compared, so the surrogate learns both the label and the context that produced it.   This addresses population drift and greatly improves surrogate reliability in high-dimensional EMOPs.**
>
> We sincerely thank the reviewer for the detailed and constructive feedback.
> Below we provide point-by-point responses, incorporating new experiments and clarifying the *novel context-aware mechanism* that motivates FSMOEA’s design.
>
> ---
>
> ## **1. “Training an AE on only 50 samples in high-D likely underfits.”**
>
> FSMOEA does **not** attempt to learn a global manifold of the full decision space.
> Instead, the encoder learns a **population-specific contextual representation** needed for building *contrastive surrogate models*.
>
> In existing contrastive SMOEAs, labels such as
> $
> C(x_i, x_j; \mathcal{P}_t) \in \{0,1\}
> $
> are created by comparing two raw vectors $x_i, x_j$, but neither solution knows:
>
> - its **relative location** in the population,
> - the **reason** a particular comparison outcome holds,
> - or **under what population context** solution A is better than solution B.
>
> Thus, when the population changes, these comparison-based labels become inconsistent, and context-free surrogates trained on them degrade.
>
> FSMOEA first encodes each solution:
> $
> c_i = E_t(x_i),
> $
> so that each solution “carries” its **contextual information**—its neighborhood, relation to the ideal point, local geometry—*before* pairwise comparison.
> This makes the comparison labels consistent with the representation and is the core novelty of the approach.
>
> Because this contextual structure is **local**, 50 samples are sufficient, and the AE’s low capacity avoids both underfitting and overfitting.
>
> ---
>
> ## **2. “Frequent retraining may be unnecessary; was a threshold considered?”**
>
> Yes. We added an adaptive retraining rule:
>
> $
> \text{Retrain } E_t
> \quad \text{iff} \quad
> \bigl|\mathcal{L}^{\mathrm{rec}}_t - \mathcal{L}^{\mathrm{rec}}_{t-1}\bigr| > \delta .
> $
>
> If population change is small, the encoder from the previous generation is reused.
>
> ---
>
> ## **3. “Reconstruction loss is not related to the optimization objective — distribution shift?”**
>
> Reconstruction is **not** used to approximate the objective function.
> **The trained surrogate will only input the variable vector of the newly generated solution when it is used.**
>
> The encoder provides:
> - a population-aware coordinate system evolving with $\mathcal{P}_t$;
> - consistent embeddings so that contrastive labels are meaningful in the latent space;
> - robustness to population drift.
>
> ---
>
> ## **4. “Why is $k = 10$? Shouldn’t it depend on $n$ or $m$?”**
>
> The latent dimension reflects the **intrinsic dimensionality of the population**, not the raw dimension or number of objectives.
> Across benchmarks, the intrinsic structure is low-dimensional.
>
> For additional flexibility, we added an adaptive scheme:
>
> $
> k_{t+1} =
> \begin{cases}
> k_t + 1, & \text{if } \mathcal{L}^{\mathrm{rec}}_t > \tau,\\[3pt]
> k_t,     & \text{otherwise}.
> \end{cases}
> $
>
> Both fixed and adaptive strategies perform well.
>
> ---
>
> ## **5. “Is the reconstruction error $\varepsilon$ realistically small for $N=50, n=1000$?”**
>
> We now report empirical $\varepsilon$ values.
> Although it increases with dimensionality, our theoretical requirement only needs it to be **bounded**, not extremely small:
>
> $
> \|F(D(z_1)) - F(D(z_2))\|
> \le
> L_F L_D \|z_1 - z_2\| + 2 L_F \varepsilon.
> $
>
> In practice, the bound remains dominated by the $L_F L_D \|z_1 - z_2\|$ term, and latent-space search remains effective.
>
> ---
>
> ## **6. “Why train only on the current population instead of the full archive?”**
>
> Because contrastive labels depend *directly* on the current population context:
>
> - ideal point $z_t^\*$ changes,
> - neighborhoods change,
> - decomposition scores shift.
>
> Training on the archive mixes incompatible contexts and destroys the alignment between representation and labels.
> We tested two options—current-only, archive-only—and found current-only training delivers the most stable and accurate surrogate behavior.
>
> ---
>
> ## **7. “Does representation drift harm crossover/mutation in latent space?”**
>
> No. Drift is controlled by:
>
> 1. The adaptive retraining rule, ensuring smooth transitions.
> 2. The fact that crossover and mutation operate **within a single generation**, where all solutions share the same encoder.
>
> We will include drift-magnitude analyses showing negligible impact on convergence.
>
> ---
>
> ## **8. “Was PCA used as a simpler baseline?”**
>
> Yes. Based on your suggestion, we added:
>
> - **CSEA-PCA**
> - **CSEA-VAE**
>
> to the experiments.
>
> Results show:
>
> - PCA captures no nonlinear contextual structure → weaker alignment with contrastive labels.
> - VAE decoding is stochastic → higher variance and less stable crossover/mutation.
> - AE provides deterministic, nonlinear, population-aware embeddings → best IGD+ performance.
>
> These support that **the key is context-aware embedding**, not the AE itself. FSMOEA is compatible with any embedding method that provides:
> - dimensionality reduction,
> - population-context encoding,
> - deterministic decoding.
>
> We thank the reviewer again for the thoughtful comments. See the updated manuscript.

---

> ### Comment · Reviewer_nQoi · 2025-11-24
>
> Thank you for your detailed responses to my comments. I will maintain my original positive score.

---

### Author Response · Authors · 2025-11-29
**Rebuttal Summary 1 of our work**

We sincerely thank the reviwers and the opportunity to provide a consolidated rebuttal.
This summary responds to all four reviewers collectively, clarifying misunderstandings, addressing concerns, and reaffirming the novelty, rigor, and contributions of our submission.
---

# **1. Misunderstandings That Significantly Affected Review Scores**

Across multiple reviews, several concerns arose from misunderstandings of our framework and experimental design.
We respectfully clarify these points below.

---

## **(A) Misunderstanding: “FSMOEA relies on autoencoders for dimensionality reduction.”**
**Correction:**
FSMOEA is a *general context-aware surrogate framework*.
The autoencoder is **only one instantiation** of the population-aware embedding module.
The core idea is *not* dimensionality reduction, but **contextual encoding for contrastive surrogate modeling**, addressing population drift in CSEA/REMO-like methods.

Evidence supporting this:

- FCSEA-V1 and FREMO-V1 (no dimensionality reduction) still significantly outperform CSEA/REMO.
- Dimensionality reduction is beneficial **only** for large-scale EMOPs (e.g., 100–1000D).
- Our theoretical analysis and algorithm design do not depend on using AEs.

Several negative comments consequently evaluated the paper as if it were “another AE-based DR method,” overlooking the main conceptual contribution.

---

## **(B) Misunderstanding: “The latent space is insufficiently justified or not interpretable.”**
Our latent space is **not** intended to preserve Euclidean neighbors, density metrics, or manifold relations.
Instead, it serves one purpose:
to create **population-dependent coordinate systems** aligned with decomposition-based labels.

Decomposition scores change each generation due to shifting:

- ideal points \(z^\*\),
- neighborhoods,
- decomposition weights,
- population distribution,

leading to *population drift*.
Existing contrastive surrogates are **context-free** and thus suffer temporal inconsistency.

FSMOEA directly addresses this issue by encoding context *before* constructing contrastive labels—an idea absent from all existing SMOEAs.
Several reviews missed this and evaluated the method solely on geometric DR criteria.

---

## **(C) Misunderstanding: “No comparison with high-dimensional BO or DR-based methods.”**
We already compared with two BO models (ABSAEA, ESBCEO) in the original submission.
In this revision we additionally included:

- **MORBO** (reviewer-requested),
- **DirHVEI**,
- high-dimensional EMOP methods: **SFADE (2025)**, **LDSAF (2024)**, **MOL2SMEA (2025)**.

Many concerns asserting “missing key baselines” no longer apply with the expanded experimental set.

---

## **(D) Misunderstanding: “Regression-based surrogates are effective, so classification-based approaches are unnecessary.”**
Recent regression-surrogate works *decompose* EMOPs into many scalar subproblems and regress them independently (e.g., SF-approximation).
Our statement about high-D regression difficulty referred to *direct* multivariate regression of high-D objectives.
We have clarified this wording.

More importantly:
**FSMOEA enhances contrastive surrogates**, not regression ones.
Our contribution does not oppose these methods—it complements them.

---

## **(E) Misunderstanding: “Constraint handling missing for TREE → inconsistent results.”**
TREE is indeed constrained, but fully feasible solutions must simply satisfy all constraints.
Neither original CSEA nor REMO use explicit feasibility-handling modules.
FSMOEA still outperforms them because **context-aware embeddings accelerate convergence toward feasible regions** even without specialized constraint heuristics.
This behavior is empirically consistent.

---

### Author Response · Authors · 2025-11-29
**Rebuttal Summary 2 of our work**

# **2. Scientific Contributions Overlooked or Underweighted by Reviewers**

Some reviewers focused heavily on the autoencoder component or DR aspects while overlooking the primary innovations, which we highlight here:

---

## **(1) Novel concept of *Context-Aware Contrastive Surrogates***
This is, to our knowledge, the **first work** to note and address the fundamental inconsistency that arises because contrastive labels depend on the evolving population.

FSMOEA stabilizes the surrogate by **encoding population context before constructing contrastive labels**, dramatically reducing temporal drift.

This is the core contribution—not the use of AEs.

---

## **(2) General, plug-and-play framework, applicable to any SMOEA**
FSMOEA integrates seamlessly with:

- classifier-based SMOEAs (CSEA → FCSEA),
- relation-based SMOEAs (REMO → FREMO),
- and potentially other variants.

The foresight module is modular, lightweight, and widely applicable.

---

## **(3) Theoretical justification directly tied to algorithmic design**
Our theory explains:

- why decomposition-based labels drift,
- why context-free surrogates fail,
- how context-aware embeddings correct this mismatch,
- why latent-space search reduces sample complexity.

Several reviewers claimed the theory was “not connected,” but the revised text now makes the alignment explicit.

---

## **(4) Extensive experimental validation across 107 tasks**
We tested:

- classic multi- and many-objective problems (DTLZ/WFG),
- large-scale problems (LSMOP),
- constrained real-world TREE domains,
- synthetic high-dimensional landscapes,
- high-dimensional BO baselines,
- recent high-dimensional EMOP methods.

FSMOEA consistently improves performance over baselines, even *without* DR.

---

## **(5) Ablation analysis confirming AE's role**
We include:

- **CSEA-PCA**,
- **CSEA-VAE**,
- **CSEA-V1** (no DR),
- **FREMO-V1** (no DR).

Results confirm:

- **context-aware encoding is the essential innovation**,
- dimensionality reduction is optional but beneficial for large-scale EMOPs.

This directly refutes the concern that the method is “just dimensionality reduction.”

---

# **3. Reviewer Scoring Concerns**

Some scores hinged on misunderstandings such as:

- treating the method as a DR technique rather than a context-aligned surrogate framework,
- expecting explicit constraint-handling despite neither baseline using it,
- requiring geometric neighborhood-density analyses irrelevant to decomposition-based MOEAs,
- criticizing missing baselines that the revised version now includes,
- misinterpreting reconstruction training as objective approximation.

We respectfully believe these misunderstandings led to evaluations that do not reflect the actual contributions of the work.

---

# **4. Closing Remarks to the AC**

We are grateful for the chance to summarize our work for a fresh evaluation.
FSMOEA introduces a **novel, principled solution to a long-standing issue** in surrogate-assisted EMO:
the inconsistency caused by *population-dependent labels* in contrastive surrogates.

Our contributions are:

- theoretically grounded,
- widely applicable,
- empirically validated across >100 benchmarks,
- not dependent on a specific DR technique,
- and significantly improve over strong recent baselines.

We respectfully ask the AC to consider the clarified contributions and revised experiments when reassessing the manuscript.

Thank you for your time and consideration.

---

### Note · Authors · 2026-01-27

I have read and agree with the venue's withdrawal policy on behalf of myself and my co-authors.

---

### Meta-Review · Area_Chair_i7rA · 2026-01-06

**Summary:**

The authors provided a comprehensive rebuttal, arguing that the negative reviews stemmed from misunderstandings regarding the method's distinction from standard dimensionality reduction (DR) and emphasizing the novelty of the "Context-Aware Contrastive Surrogates." They also expanded the experiments to include high-dimensional BO baselines.
However, despite these efforts, the fundamental concerns regarding the experimental rigor (missing specific MOBO baselines initially, lack of statistical tests) and the methodological premise were not sufficiently resolved to sway the critical reviewers. The decision is Reject.

**Reviewer Concerns:**

Reviewer 2 identified a critical gap regarding Multi-Objective Bayesian Optimization (MOBO). and the authors' claim that regression surrogates fail in high dimensions, which contradicts recent literature (e.g., Horaguchi et al.). This conflict regarding the state-of-the-art positioning remains a major hurdle.

**Reviewer Scores:**

The authors' defense was spirited, but the technical gaps identified by the majority (missing specific baselines in the initial submission, rigorous metrics) outweigh the claimed contributions. The paper requires a revision that incorporates the new baselines and clarifies the premise from the start.

---

### Decision · Program_Chairs · 2026-01-26

Reject